# Reconciling Model Multiplicity for Downstream Decision Making

**Ally Yalei Du***
Carnegie Mellon University
aydu@andrew.cmu.edu

**Dung Daniel Ngo***
University of Minnesota
ngo00054@umn.edu

**Zhiwei Steven Wu**
Carnegie Mellon University
zstevenwu@cmu.edu

## Abstract

We consider the problem of *model multiplicity* in downstream decision-making, a setting where two predictive models of equivalent accuracy cannot agree on what action to take for a downstream decision-making problem. Prior work attempts to address model multiplicity by resolving prediction disagreement between models. However, we show that even when the two predictive models approximately agree on their individual predictions almost everywhere, these models can lead the downstream decision-maker to take actions with substantially higher losses. We address this issue by proposing a framework that *calibrates* the predictive models with respect to both a finite set of downstream decision-making problems and the individual probability prediction. Specifically, leveraging tools from multi-calibration, we provide an algorithm that, at each time-step, first reconciles the differences in individual probability prediction, then calibrates the updated models such that they are indistinguishable from the true probability distribution to the decision-makers. We extend our results to the setting where one does not have direct access to the true probability distribution and instead relies on a set of i.i.d data to be the empirical distribution. Furthermore, we generalize our results to the settings where one has more than two predictive models and an infinitely large downstream action set. Finally, we provide a set of experiments to evaluate our methods empirically. Compared to existing work, our proposed algorithm creates a pair of predictive models with improved downstream decision-making losses and agrees on their best-response actions almost everywhere.

## 1 Introduction

In many applications, individual probability prediction is central to decision-making. For example, in the Job Training Partnership Act (JTPA) training program (Bloom et al., 1997), a decision maker may predict whether an individual is employed before assigning them to training; or in medical trials, a doctor may predict the probability that the patient has contracted a disease before recommending them a treatment. Since the hospital does not know the true probability that a particular patient is ill, they can only evaluate the individual probability predictions through its average outcome over a sufficiently large sample set. For a predictive task, the standard convention is to choose the model that maximizes *accuracy*. However, previous work has shown that it is common to have multiple predictive models with similar accuracy but substantially different properties (Chen et al., 2018; Rodolfa et al., 2020; D'Amour et al., 2022). This phenomenon is called *predictive* (or model) multiplicity, a line of work studied by Breiman (2001); Marx et al. (2020); Black et al. (2022).

In predictive multiplicity scenarios, decision-makers may have multiple models with similar accuracy but conflicting predictions on individual samples. In our motivating example (Figure 1), the hospital has access to two models $f_1$ and $f_2$ predicting the probability of disease which are equally accurate on average over the entire population in terms of squared loss, but their predictions on a subpopulation may vastly differ. This disagreement in outcome prediction may have a disparate impact on the subpopulation if the hospital has to choose one predictor over the other to make important downstream decisions. For example, they might select a treatment based on the predicted probability that a patient has contracted a disease. Formally, given a predictive model $f$ and a decision-making loss function $\ell(y, \cdot)$, the decision-maker wants to choose a best-response action, i.e., the action $a$ that minimizes $\mathbb{E}_{y \sim f}[\ell(', a)]$. When two models $f_1$ and $f_2$ have nearly equivalent

accuracy but lead to different best-response actions, the decision-maker would not be able to identify which best-response action to take for individual patients. While predictive multiplicity offers great flexibility for the decision-maker in the model selection process, it also places an additional burden on the decision-maker to correctly navigate such freedom and justify how they use a predictive model to make downstream decisions.

Roth et al. (2023) address model multiplicity by resolving prediction disagreements using techniques from multi-calibration (Hebert-Johnson et al., 2018), which aligns the mean of estimated probability with data across diverse reference groups. Specifically, Roth et al. (2023) provide a procedure called "Reconcile" that updates the predictive models to minimize their disagreements and improve the accuracy of each model. However, we show simple settings where the reconciled predictions from Roth et al. (2023) can lead the downstream decision-makers to take actions with substantially higher losses. We visually demonstrate this scenario in Figure 1. For a more detailed discussion on the limitation of prior work, see Section 2.4 and Appendix B. This motivates the study of how to reconcile predictive multiplicity with an explicit focus on its impact on downstream decisions.

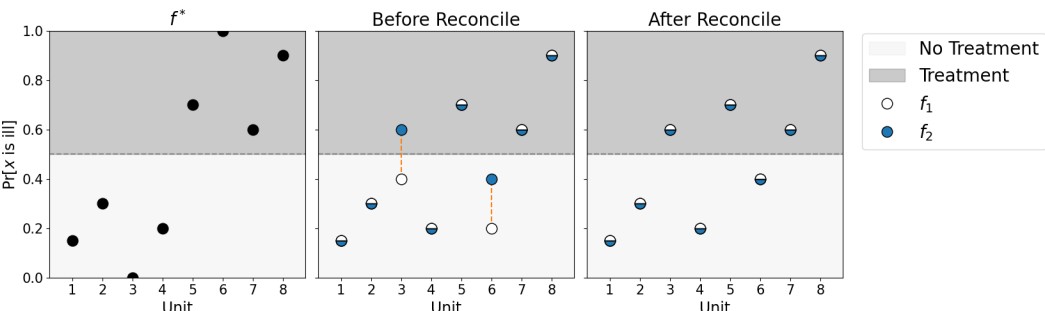

Figure 1: An illustrative example of the drawback in a prior work's attempt at addressing model multiplicity. Consider a stylized binary classification problem on a dataset with $8$ units (patients) and the hospital deciding between two actions (treatment vs. no treatment). Treatment is assigned if the predicted probability is above $1/2$. **Left**: The true probability that each patient is labeled 'ill'. **Middle**: The predicted probability that each patient is ill according to $f_1$ (white) and $f_2$ (blue). While these two predictors have almost the same accuracy (in squared loss), their individual probability predictions for patients $3$ and $6$ vastly differ. **Right**: After running the Reconcile procedure of Roth et al. (2023), the individual probability predictions agree everywhere. However, the best-response action of unit $3$ changed from correct (no treatment) to incorrect (treatment). If the hospital uses the updated $f_1$ to make their treatment recommendation, they would incur more loss than before had they not updated the predictor using Reconcile. This example is formalized in Theorem 2.7.

In this work, our goal is to leverage tools from multi-calibration (Hebert-Johnson et al., 2018) to alleviate the model multiplicity issue in high-dimensional decision-making tasks for multiple decision-makers with multiple decision-making loss functions. Specifically, we show how the decision-maker can update the input predictors so they approximately agree on (1) individual predictions and (2) best-response actions for each individual in the downstream decision-making task. Our procedure ensures that the number of disagreements in best-response actions decreases over time, which enables the decision-makers to confidently use either of the updated predictors to justify their decisions.

**Overview of Paper.** We study the problem of reconciling model multiplicity for multiple downstream decision-making tasks, where the decision-makers have multiple predictive models with nearly equivalent squared loss but may lead to vastly different best-response actions for a significant number of individuals in the population. Our key contributions are summarized as follows.

- In Section 2, we formulate the problem of model multiplicity from the perspective of the decision-makers. In Section 2.4, we formalize our motivating example in Figure 1 and show that it is insufficient to only update two predictive models so that they have improved squared loss and nearly agree on their individual predictions almost everywhere.
- In Section 3, we introduce an algorithm, ReDCal, that outputs predictive models that are (1) calibrated to a finite set of downstream decision-making tasks and (2) approximately agree on their predictions and best-response actions almost everywhere for each downstream task.

- In Section 3.2, we extend our analysis to the setting where one does not have direct access to the true distribution and instead only has a validation dataset with samples drawn i.i.d from the underlying distribution. We show that the guarantees obtained using the empirical distribution can be translated to the unknown underlying distribution.

- In Section 3.3, we extend our analysis to the setting where one has $k > 2$ predictors that need to be reconciled. We provide an efficient "contestation" algorithm that only requires approximately $k$ times more samples than the base model that reconciles 2 predictors. In Appendix F, we further generalize our result to the setting where the action space of the downstream decision-making tasks is infinitely large.

- Finally, in Section 4, we empirically evaluate the performance of the proposed algorithm on real-world datasets and show our improvement over the benchmark prior work in resolving disagreement in downstream decision-making tasks.

## 1.1 RELATED WORK

**Model multiplicity.** Within the literature on predictive multiplicity, our work builds off the line of work focusing on predicting individual probabilities (Marx et al., 2020; D'Amour et al., 2022; Black et al., 2022; Breiman, 2001), where solving an error minimization problem for some prediction tasks can lead to multiple solutions with roughly similar performance in terms of accuracy. Sandroni (2003) showed that one cannot empirically distinguish the outcomes from a predictor encoding the true individual probabilities from one without in isolation, while Al-Najjar & Weinstein (2008); Feinberg & Stewart (2008) provided comparative tests to differentiate between the true probability predictor and one that is not. Particularly, Feinberg & Stewart (2008) relied on *cross-calibration*, i.e., calibration conditional on the predictions of both models to empirically falsify one of them. For downstream decision-making, Garg et al. (2019) worked on refining predictors and provided an algorithm that produces a predictor $f_3$ that is cross-calibrated with respect to both $f_1$ and $f_2$. An alternative framework studied by Globus-Harris et al. (2022b) seeks to update models that are suboptimal for different subsets of the population, following the 'bug bounties' approach used by the software and security communities. Towards the same goal, Roth et al. (2023) proposed an algorithm ("Reconcile") that aims to reconcile different predictors with equivalent errors such that the updated predictors both have lower errors compared to the initial models and approximately agree on their prediction on almost all units. Our work follows the reconciliation idea from Roth et al. (2023) but focuses on not only the predictive accuracy but also the downstream decision-making. In our model, we consider reconciling predictors for both regression and multi-class classification problems, and their impact on the downstream decision-making tasks. In Section 2.4, we provide a numerical example where simply reconciling the probability predictions according to Roth et al. (2023) can lead to additional losses in downstream decision-making tasks.

**Multi-calibration.** Our work draws on techniques from the growing literature on multi-calibration (Hebert-Johnson et al., 2018; Kim et al., 2019; Dwork et al., 2019; Shabat et al., 2020; Jung et al., 2020; Dwork et al., 2021; Jung et al., 2022; Haghtalab et al., 2023; Deng et al., 2023; Noarov et al., 2023; Globus-Harris et al., 2022a; 2023). Multi-calibration has been used as a notion of fairness as it guarantees calibration for any identifiable group. (A more detailed discussion of calibration and fairness is given in Appendix A.) Particularly, we leverage the connection between multi-calibration and loss minimization from the omniprediction literature (Gopalan et al., 2022b;a; 2021) in our analysis. Within the framework of multi-calibration, the work most related to ours is that of Zhao et al. (2021), who considered decision calibration with respect to all classes of loss functions. Similar to us, Zhao et al. (2021) takes the perspective of a decision-maker who wants to ensure the predictive models are *indistinguishable* from the true probability when they are used to make downstream decisions. However, two decision-calibrated models can still disagree on their individual predictions and best-response actions for many units. In our motivating example, a hospital with two decision-calibrated predictors may still aim for consistent predictions and downstream decision recommendations (i.e. treatment or not) across the population. We formalize this example in Appendix B and provide empirical experiments to show our improvement over their result in Section 4.

An independent and concurrent work by Globus-Harris et al. (2024) considers ensembling multiple predictive models for high-dimensional downstream decision-making tasks. While their work also leverages techniques from multi-calibration, their goal is to output an ensemble predictor whose

self-estimated expected payoff is accurate and whose induced policy has a payoff at least as high as the maximum self-assessed payoff of individual models. In contrast, our interest is in reducing the downstream decision losses by resolving the differences between equivalent predictors and mitigating model multiplicity for decision-making.

## 2 PROBLEM FORMULATION

**Notation.** Throughout this paper, we use subscripts $i$ to index different predictions, superscripts $t$ to index different time-steps, and $a$ to index actions. For $K \in \mathbb{N}$, we use the shorthand $[K] := \{1, 2, \cdots, K\}$. $\Delta(\mathcal{X})$ denote the set of possible distributions over $\mathcal{X}$.

We consider the prediction problem with random variables $x$ and $y$, where $x \in \mathcal{X}$ represents the features and $y \in \mathcal{Y}$ represents the labels. We focus on the regression problem in which the label domain is real-valued and bounded: $\mathcal{Y} \subset [0, 1]^d$. Our formulation also permits the multi-class classification problem by writing the label $y$'s as one-hot vectors.

We denote $\mathcal{D} \in \Delta(\mathcal{X} \times \mathcal{Y})$ as the true distribution over the pairs of features-label $(x, y)$. In practice, we will not have access to $\mathcal{D}$, and instead only know a set of $n$ data points $D$ sampled i.i.d from $\mathcal{D}$. In such case, we consider the dataset $D = \{(x_1, y_1), \cdots, (x_n, y_n)\}$ to be the *empirical distribution* over $D$, which is a discrete distribution that place uniform weight $1/n$ on each sample $(x, y) \in D$.

A predictor is a map $f : \mathcal{X} \to [0, 1]^d$. Our goal is to find the Bayes optimal predictor $f^* : \mathcal{X} \to [0, 1]^d$ such that for all $x \in \mathcal{X}$, $f^*(x) = \mathbb{E}_{(x,y) \sim \mathcal{D}}[y|x]$ is the *conditional label expectation given $x$*.

### 2.1 MODEL EVALUATION

Given a predictor $f \in [0, 1]^d$, we evaluate $f$ via its squared error, i.e., its expected deviation from the true label. We formalize this objective in the following definition.

**Definition 2.1** (Brier Score). *The squared error (also known as Brier score) of a predictor $f$ evaluated on distribution $\mathcal{D}$ is given as:*

$$B(f, \mathcal{D}) = \mathbb{E}_{(x,y) \sim \mathcal{D}}[\|f(x) - y\|_2^2]$$

*When we only have a dataset $D = \{(x_1, y_1), \cdots, (x_n, y_n)\}$, the empirical Brier score is given as:*

$$B(f, D) = \frac{1}{n} \sum_{i=1}^{n} \|f(x_i) - y_i\|_2^2$$

Note that we use the Brier score as our metric because it can be accurately estimated given access to only the samples from the distribution. Moreover, among all possible predictors, the Brier score is minimized by the Bayes optimal predictor $f^*$.

**Lemma 2.2.** *Fix any distribution $\mathcal{D}$ and let $f^*(x) = \mathbb{E}_{(x,y) \sim \mathcal{D}}[y|x]$ represent the true conditional label encoded by $\mathcal{D}$. Let $f : \mathcal{X} \to [0, 1]^d$ be any other model. Then we have $B(f^*, \mathcal{D}) \leq B(f, \mathcal{D})$.*

Hence, given two predictors $f_1$ and $f_2$, if we can verify empirically from the observable data that $B(f_1, \mathcal{D}) \leq B(f_2, \mathcal{D})$, then we can empirically falsify that $f_2$ encodes the true conditional label.

### 2.2 DOWNSTREAM DECISION-MAKING TASKS AND LOSS FUNCTIONS

Beyond our initial goal of finding a good estimate for the true conditional predictor $f^*$, we are also interested in using our predictors for downstream decision-making problems. Formally, we consider a loss minimization problem, where the decision-maker has a set of possible actions $\mathcal{A}$ and a loss function $\ell : \mathcal{Y} \times \mathcal{A} \to [0, d]$. Wlog, we only consider finite action set $\mathcal{A} = [A]$, i.e., there are $A$ possible actions[1]. In this paper, we assume that the loss function does not directly depend on the features $x$ and is linear in $y$. That is, for each action $a \in \mathcal{A}$, there exists some $\ell_a \in [0, 1]^d$ such that

$$\ell(y, a) = \langle y, \ell_a \rangle \tag{1}$$

---

[1] Our technical results can be generalized to a setting with infinitely many actions, i.e., $|\mathcal{A}| = \infty$. For details, see Appendix F.

We write $\mathcal{L} = \{\ell : \mathcal{Y} \times \mathcal{A} \to [0, d]\}$ to denote a finite family of loss functions. We consider the setting with multiple different decision-makers, each using a different linear loss function in $\mathcal{L}$. For any loss function $\ell : \mathcal{Y} \times \mathcal{A} \to \mathbb{R}$, we can rescale each coordinate of $\ell_a$ to be between $[0, 1]$.

Given a predictor $f$ and a loss function $\ell$, the decision-maker selects an action $a \in \mathcal{A}$ that minimizes the expected loss. We define the best-response policy taken by the decision-maker as follows.

**Definition 2.3** (Best-response policy). *Given a loss function $\ell$, a predictor $f$ and the action set $\mathcal{A}$, the best-response policy for $\ell$ is given as*

$$\pi_\ell^{\mathsf{BR}}(f(x)) = \arg\min_{a \in \mathcal{A}} \langle f(x), \ell_a \rangle.$$

### 2.3 CALIBRATION

In our setting, we consider the decision-maker only having access to some pre-trained predictors $f$ given by a third-party. For instance, a data scientist trained a pair of models on an image dataset without exact knowledge of how the downstream decision-maker will use such predictors. We may imagine the decision-maker as a hospital considering whether to recommend treatment to certain patients based on the predicted probability that the patient has contracted a skin disease. Since the hospital's treatment-recommendation algorithm is not known to the public (and the data scientist), we assume that the data scientists initially aim to minimize the squared error in their predictions.

Since the hospital believes that the input predictors may not perform well according to their own loss function, they want the data scientist to convey trust through other performance guarantees of the predictors. One such guarantee is multi-calibration with respect to a finite set of loss functions $\mathcal{L} \ni \ell$ and a set of events $\mathcal{E}$ on the best-response policy, i.e., if the loss function $\ell$ belongs to $\mathcal{L}$, the decision-maker should be able to accurately compute the expected loss of choosing an action using the best-response policy $\pi_\ell^{\mathsf{BR}}$. Formally, we let $E_{\ell,a}(f(x), x)$ denote the action selection events:

**Definition 2.4** (Best-response Events). *Given a predictor $f$, let $\mathcal{E}$ be the set of best-response events, where each event $E_{\ell,a} \in \mathcal{E}$ for some loss function $\ell$ and action $a \in \mathcal{A}$ is defined as*

$$E_{\ell,a}(f(x), x) = \mathbf{1}[x \in \{x : \pi_\ell^{\mathsf{BR}}(f(x)) = a\}]$$

Given a set of events $\mathcal{E}$, we can define an approximate notion of multi-calibration with respect to $\mathcal{E}$.

**Definition 2.5** ($\beta$-approximate decision calibration). *A predictor $f$ is $\beta-$decision calibrated with respect to the set of best-response events $\mathcal{E}$ if for all $E_{\ell,a} \in \mathcal{E}$, we have:*

$$\left\| \mathbb{E}_{(x,y) \sim \mathcal{D}}[(y - f(x)) \cdot E_{\ell,a}(f(x), x)] \right\|_2 \leq \beta.$$

This definition follows from an equivalent definition of decision calibration in Zhao et al. (2021). The main difference is we define calibration with respect to a set of events on the best-response actions following the formulation of multi-calibration for online learning in Noarov et al. (2023) and a generalization of multi-calibration in Deng et al. (2023). This definition implies that if a predictor $f$ is $\beta-$decision calibrated with respect to the best-response events $\mathcal{E}$, then the decision-maker can accurately estimate the expected loss from using $f$ to make decisions.

**Lemma 2.6** ((Zhao et al., 2021)). *For all $a, a' \in \mathcal{A}, \ell \in \mathcal{L}$, if $f$ is $\beta$-decision-calibrated with respect to the best-response events $\mathcal{E}$, then the loss estimation satisfies*

$$\left| \mathbb{E}_{(x,y) \sim \mathcal{D}}[\ell(y, a') \cdot E_{\ell,a}(f(x), x)] - \mathbb{E}_{x \sim \mathcal{D}_{\mathcal{X}}}[\langle f(x), \ell_{a'} \rangle \cdot E_{\ell,a}(f(x), x)] \right| \leq \beta \sqrt{d}$$

### 2.4 LIMITATIONS OF PRIOR WORKS

In this section, we show that, for the predictive model multiplicity problem under decision-making, improving the accuracy until the two predictors agree on their predictions almost everywhere is not a sufficient solution to our problem. In our analysis below, we consider a stylized problem with $\mathcal{Y} = \{0, 1\}$ and $\mathcal{A} = \{0, 1\}$, i.e., binary class and binary action space. A predictor here is $f : \mathcal{X} \to [0, 1]$, and the optimal predictor is $f^*(x) = \Pr_{(x',y') \sim \mathcal{D}}[y' = 1 | x' = x]$. As a shorthand, we denote $f_1(1)$ as the probability of unit 1 being labelled 1. The loss is defined as

$$\ell(0, 0) = \ell(1, 1) = 0, \quad \ell(1, 0) = \ell(0, 1) = 1, \tag{2}$$

That is, for any $x$, the best-response policy is to take action 0 if $f(x) \leq 1/2$ and action 1 otherwise.

**Reconcile individual predictions.** Prior work by Roth et al. (2023) considers the model multiplicity problem for individual probability predictions. Their proposed algorithm, Reconcile (Algorithm 3), iteratively adjusts predictions for subsets where models differ significantly, aligning predicted and true mean values. The algorithm ultimately returns a pair of predictors that has a smaller Brier score than the input predictors and approximately agree on their predictions on almost all units. In the following theorem, we show that the best-response policy induced by the predictors updated by Reconcile might lead to a higher expected loss than the ones they started with.

**Theorem 2.7.** *For any $\alpha \in (0, 1/3)$, $\eta \in (0, 1)$, there exists a pair of predictors $f_1, f_2$ such that after running Algorithm 3 (Reconcile), the output models $f_1^T, f_2^T$ satisfy $f_1^T(x) = f_2^T(x), \forall x \in \mathcal{X}$, but there exists a loss function $\ell$ such that at least one of $f_1^T, f_2^T$ induce worse losses compared to the original models $f_1, f_2$. That is, for some $i \in [2]$,*

$$E_{(x,y)\sim\mathcal{D}}[\ell(y, \pi^{\mathsf{BR}}(f_i^T(x))) - \ell(y, \pi^{\mathsf{BR}}(f_i(x)))] = 1/2 > 0 \qquad (3)$$

*Proof.* Consider a setting with $\mathcal{X} = [2]$ and $\Pr[x = 1] = \Pr[x = 2] = 0.5$. For any $0 < \alpha < 1/3$, let $\phi \geq \alpha$, we consider the two predictors $f_1, f_2$ defined as follows:

$$f_1(1) = 1/2 - \phi/2, \quad f_1(2) = 1/2 - 3\phi/2, \quad f_2(1) = 1/2 + \phi/2, \quad f_2(2) = 1/2 - \phi/2 \qquad (4)$$

and the true probability of each unit being labeled 1 are $f^*(1) = 0$ and $f^*(2) = 1$.

The Brier scores of $f_1$ and $f_2$ differ only by $\phi^2$, but their individual predictions differ for both features $x = 1$ and $x = 2$. We can run Algorithm 3 and patch $f_1$ to get the updated model $f_1^T$ with

$$f_1^T(1) = 1/2 + \phi/2 = f_2(1), \quad f_1^T(2) = 1/2 - \phi/2 = f_2(2).$$

Consider the loss function $\ell$ defined in equation 2. The change in expected loss after patching $f_1$ is

$$\mathbb{E}_{(x,y)\sim\mathcal{D}}[\ell(y, \pi_\ell^{\mathsf{BR}}(f_1^T(x))) - \ell(y, \pi_\ell^{\mathsf{BR}}(f_1(x)))] = 1/2 > 0.$$

Therefore, no matter how small $\alpha$ and $\eta$ are, the loss of predictor $f_1$ increases by a constant amount. $\square$

Moreover, we provide a counterexample to show that it is insufficient to only ensure each individual predictor is approximately decision-calibrated using Algorithm 1. See Appendix B for details.

## 3 RECONCILE FOR DECISION MAKING

Suppose we are given two predictors $f_1, f_2 : \mathcal{X} \to [0, 1]^d$, where $f_1, f_2$ have nearly equivalent accuracy in terms of Brier score differ in their induced decision-making policies, but we cannot falsify either of the two from the data. Informally, our goal is to return a pair of models $f_1', f_2'$ such that: (1) for both $i \in \{1, 2\}$, $f_i'$ is more accurate than $f_i$ in terms of Brier score; (2) for both $i \in \{1, 2\}$, the best-response policy induced by $f_i'$ has no larger expected loss than that of $f_i$; (3) $f_1'$ and $f_2'$ approximately agree almost everywhere, indicating limited room for additional improvement.

To this end, we are interested in the region where the two predictors disagree substantially with respect to the downstream decision-making task. We define the disagreement region as follows:

**Definition 3.1** (Disagreement Event). *For $f_1, f_2$, margin $\alpha > 0$, and a loss function $\ell$, the disagreement event is defined for a pair of best-response actions $a_1, a_2 \in \mathcal{A}$ where $a_1 \neq a_2$ as*

$$E_{\ell,a_1,a_2}^\alpha(f_1(x), f_2(x), x) = \mathbf{1}\Big[x \in \big\{x : \pi_\ell^{\mathsf{BR}}(f_1(x)) = a_1, \pi_\ell^{\mathsf{BR}}(f_2(x)) = a_2,$$

$$\langle f_1(x), \ell_{a_2} - \ell_{a_1}\rangle > \alpha \text{ or } \langle f_2(x), \ell_{a_1} - \ell_{a_2}\rangle > \alpha\big\}\Big],$$

As shorthand, we denote $E_{\ell,a_1,a_2}(x) = E_{\ell,a_1,a_2}^\alpha(f_1(x), f_2(x), x)$ when the predictors $f_1, f_2$ and the margin $\alpha$ are clear from context. For a finite family of loss functions $\mathcal{L}$, we can always iterate through $\mathcal{L}$ to identify the tuple $(\ell, a_1, a_2)$ that defines a disagreement region between $f_1$ and $f_2$.

We say the two models approximately agree with each other when the size of the disagreement event is small enough, i.e., its probability mass $\mu(E_{\ell,a_1,a_2})$ on the underlying distribution $\mathcal{D}$ is small.

**Definition 3.2.** *(Probability Mass) Under distribution $\mathcal{D}$, the probability mass $\mu(E)$ of event $E$ is*

$$\mu(E) = \Pr_{(x,y)\sim\mathcal{D}}[E(x) = 1].$$

## 3.1 THE RECONCILE PROCEDURE

In this section, we propose our main algorithm ReDCal (Algorithm 2). Whenever the decision-maker observes a large disagreement event $E_{\ell,a_1,a_2}$, the best-response action and its corresponding expected loss given by at least one of the predictors must be incorrect. For example, at time step $t$ and unit $x$, if the gap between the losses of taking $a_1$ and $a_2$ according to $f_2$ is substantially different from the loss gap observed on the data, then the decision-maker can induce that $f_2$ must have been wrong in its prediction for $x$. Then, the decision-maker would want to 'patch' $f_2$ in this time-step.

The calibration procedure within each time-step is divided into two stages. In the first stage, we update model $f_2$ to $f_2'$ by minimizing the mean prediction error on the disagreement event, i.e., minimizing $\mathbb{E}[\|f_2'(x) - y|E_{\ell,a_1,a_2}(x) = 1\|]$. Following the intuition from multi-calibration, this update operation would improve the Brier score and produce a more accurate predictor. However, the updated model $f_2'$ is not guaranteed to induce the correct best-response action and could instead induce some other actions that might lead to a larger expected loss. To cope with this, in the second stage, we further update $f_2'$ to a model $f_2''$ that is approximately decision-calibrated within event $E_{\ell,a_1,a_2}$ using Algorithm 1. Since the loss estimation given by $f_2''$ is accurate for all best-response events within $E_{\ell,a_1,a_2}$ and we are taking actions to minimize estimated loss, we can now safely take the best-response action induced by $f_2''$. The formal description of the algorithm is given by Algorithm 2.

---

**Algorithm 1:** Decision Calibration

**Input:** Predictor $f$, loss family $\mathcal{L}$, $\beta > 0$, event $E$
1: Let $f^0 = f$.
2: **while** $f^t$ is not $\beta$-multicalibrated with respect to events $E_{\ell,a} \cap E$ for some $\ell \in \mathcal{L}$ **do**
3:    Let $\ell^t, a^t = \arg\max_{\ell,a} \|\mathbb{E}_{(x,y)\sim\mathcal{D}}[(y - f^t(x))E_{\ell,a}(f^t(x), x)]\|_2$
4:    Let $\phi^t = \mathbb{E}_{(x,y)\sim\mathcal{D}}[y - f^t(x)|E_{\ell^t,a^t}(f^t(x), x) = 1]$
5:    Patch $f^{t+1}(x) = \text{proj}_{[0,1]^d}(f^t(x) + \phi^t E_{\ell^t,a^t}(f^t(x), x))$
6:    $t = t + 1$.
**Output:** $f^t$

---

**Algorithm 2:** Reconcile Decision Calibration (ReDCal)

**Input:** $f_1, f_2, \mathcal{L}, \eta > 0, \alpha > 0, \beta > 0$
1: Let $f_1^0 = f_1, f_2^0 = f_2$ and $t = 0$.
2: **while** $\mu(E_{\ell,a_1,a_2}) \geq \eta$ for some $a_1, a_2 \in \mathcal{A}$ and $\ell \in \mathcal{L}$ **do**
3:    Let $\ell^t, a_1^t, a_2^t = \arg\max_{\ell,a,a'} \mu(E_{\ell,a,a'})$, $E^t = E_{\ell^t,a_1^t,a_2^t}$.
4:    Pick

$$i^t = \arg\max_{i \in \{1,2\}} \big|\mathbb{E}_{(x,y)\sim\mathcal{D}}[\ell^t(y, a_1^t) - \ell^t(y, a_2^t)|E^t(x) = 1]$$
$$- \mathbb{E}_{x\sim\mathcal{X}}[\ell^t(f_i(x), a_1^t) - \ell^t(f_i(x), a_2^t)|E^t(x) = 1]\big|.$$

5:    Denote $f_{i^t}^t$ as $f_i^t$. Let $\phi^t = \mathbb{E}_{(x,y)\sim\mathcal{D}}[y|E^t(x) = 1] - \mathbb{E}_{x\sim\mathcal{D}_\mathcal{X}}[f_i^t(x)|E^t(x) = 1]$.
6:    Patch $f^t(x) = \text{proj}_{[0,1]^d}(f_i^t(x) + \phi^t E^t(x))$.
7:    Let $f_i^{t+1} = \text{Decision-Calibration}(f^t, \mathcal{L}, \beta, E^t)$. $t = t + 1$.
**Output:** $f_1^t, f_2^t$

---

We provide the theoretical guarantees of our proposed algorithm below. At a high level, Algorithm 2 produces a pair of models with improved accuracy and approximately agrees on the best-response action almost everywhere. Unlike prior work in model multiplicity, our theoretical guarantees in this section (and their extensions in later sections) do not require the input predictors $f_1, f_2$ to have similar accuracy. Using techniques from multi-calibration, our proposed algorithms can reconcile predictors with distinct accuracy, which could be beneficial to downstream decision-makers depending on the context. For the formal proofs of Theorem 3.3, see Appendix C.

**Theorem 3.3.** *For any pair of models $f_1, f_2 : \mathcal{X} \to [0,1]^d$, any distribution $\mathcal{D}$, family of loss functions $\mathcal{L}$ on a set of action $\mathcal{A}$ satisfying $|\mathcal{A}| = A$, any loss margin $\alpha > 0$, disagreement region*

*mass $\eta > 0$, and decision-calibration tolerance $\beta > 0$, Algorithm 2 updates $f_1$ and $f_2$ for $T_1$ and $T_2$ time-steps, respectively, and outputs a pair of models $(f_1^T, f_2^T)$, such that:*

1. *Algorithm 2 terminates within $T = T_1 + T_2 \leq \frac{4 \cdot d \cdot (B(f_1, \mathcal{D}) + B(f_2, \mathcal{D}))}{\alpha^2 \eta}$ time-steps.*

2. *The Brier scores of the final models are lower than that of the input models $(f_1, f_2)$:*

$$B(f_1^T, \mathcal{D}) \leq B(f_1, \mathcal{D}) - T_1 \cdot \alpha^2 \eta/(4d) \quad \text{and} \quad B(f_2^T, \mathcal{D}) \leq B(f_2, \mathcal{D}) - T_2 \cdot \alpha^2 \eta/(4d)$$

3. *All the downstream decision-making losses of the final models do not increase by much compared to that of the input models $(f_1, f_2)$: for each $i \in \{1, 2\}$ and for all $\ell \in \mathcal{L}$,*

$$\mathbb{E}_{(x,y)\sim\mathcal{D}}[\ell(y, \pi_\ell^{\mathsf{BR}}(f_i^T(x)))] - \mathbb{E}_{(x,y)\sim\mathcal{D}}[\ell(y, \pi_\ell^{\mathsf{BR}}(f_i(x)))] \leq T_i \beta \sqrt{d} A$$

4. *The final models approximately agree on their best-response actions almost everywhere. That is, the disagreement region $E_{\ell, a_1, a_2}$ calculated using $f_1^T, f_2^T$ has small mass. $\forall \ell \in \mathcal{L}$,*

$$\mu(E_{\ell, a_1, a_2}) < \eta \quad \text{for all } a_1, a_2 \in \mathcal{A} \quad \text{s.t } a_1 \neq a_2$$

**Remark 3.4.** *In the third result of Theorem 3.3, the increase in downstream decision-making loss at each time-step only depends on the decision-calibrate tolerance $\beta$, dimension $d$, and number of actions $A$. Since the total number of time-steps does not depend on $\beta$, we can set $\beta = \alpha/T\sqrt{d}A$ to ensure the loss of taking the best-response action does not degrade by more than $\alpha$. Moreover, in Section 4, we empirically observe that the loss only increases minimally.*

## 3.2 FINITE SAMPLE ANALYSIS

In Section 3.1, we have presented an algorithm, ReDCal, to reconcile two predictors with similar Brier scores, assuming the decision-makers have direct access to the probability distribution $\mathcal{D}$. In practice, the decision-makers will only have a dataset $D = \{(x_1, y_1), \cdots, (x_n, y_n)\}$ containing $n$ i.i.d samples drawn from $\mathcal{D}$. In this section, we will instead run Algorithm 2 on the empirical distribution over $D$ and show that its guarantees can translate to the underlying distribution $\mathcal{D}$ with high probability. To prevent data leakage, we assume that the dataset $D$ is drawn independently of the predictors $f_1$ and $f_2$, i.e., the dataset contains freshly drawn data that was not used to train either of the predictors that we want to reconcile. For the formal proofs, see Appendix D.

At a high level, since the samples in $D$ are independently and identically distributed, we can apply Chernoff-Hoeffding inequality to show that, with high probability, the in-sample quantities are approximately equal to out-sample quantities. We summarize the results in the theorem below.

**Theorem 3.5.** *Fix any distribution $\mathcal{D}$ and dataset $D$ containing $n$ samples drawn i.i.d from $\mathcal{D}$. For any pair of models $f_1, f_2 : \mathcal{X} \to [0, 1]^d$, family of loss functions $\mathcal{L}$ on a set of action $\mathcal{A}$ satisfying $|\mathcal{A}| = A$, loss margin $\alpha > 0$, disagreement region mass $\eta > 0$, and decision-calibration tolerance $\beta > 0$, Algorithm 2 run over the empirical distribution $D$ updates $f_1$ and $f_2$ for $T_1$ and $T_2$ time-steps, respectively, and outputs a pair of predictors $(f_1^T, f_2^T)$ such that, with probability at least $1 - \delta$ over the randomness of $D \sim \mathcal{D}^n$,*

1. *The total number of time-steps for Algorithm 2 and Algorithm 1 is*

$$T = T_1 + T_2 \leq 2d/\min\{\beta^2, \eta\alpha^2/4d\}$$

2. *For $i \in \{1, 2\}$, the Brier scores of the final models are lower than that of the input models:*

$$B(f_i^{T_i}, \mathcal{D}) \leq B(f_i, \mathcal{D}) - (T_i - \eta) \cdot \min\left\{\beta^2, \eta\alpha^2/(4d)\right\}$$

3. *For $i \in \{1, 2\}$ and for all $\ell \in \mathcal{L}$, the downstream decision-making losses of the final models do not increase by much compared to that of the input models:*

$$\mathbb{E}_{(x,y)\sim\mathcal{D}}[\ell(y, \pi_\ell^{\mathsf{BR}}(f_i^T(x)) - \ell(y, \pi_\ell^{\mathsf{BR}}(f_i(x))] \leq 2T_i \beta \sqrt{d} A$$

4. *The final models approximately agree on their best-response actions almost everywhere. That is, the disagreement region $E_{\ell, a_1, a_2}$ calculated using $f_1^T, f_2^T$ has small mass: $\forall \ell \in \mathcal{L}$,*

$$\mu(E_{\ell, a_1, a_2}) \leq 2\eta \quad \text{for all } a_1, a_2 \in \mathcal{A} \quad \text{s.t } a_1 \neq a_2$$

*if $n \geq \Omega\left(d^2/(\eta^2 \min\{\beta, \eta\alpha^2/d\}) \cdot \left(\ln(dA|\mathcal{L}|) - \ln(\delta \min\{\beta, \eta\alpha^2\})\right)\right)$.*

### 3.3 EXTENSION TO RECONCILING MULTIPLE PREDICTORS

In the previous sections, we addressed reconciling two predictors for downstream decision-making tasks. When there are $k > 2$ predictors that need to be reconciled, a straightforward approach to generalize our previous analysis is applying ReDCal iteratively until each pair among the $k$ predictors "agrees" on their individual prediction and best-response action. However, since there are $O(k^2)$ pairs of predictors and this approach requires fresh calibration data for each new reconciliation procedure, the data requirement significantly increases.

Another approach is to simply apply ReDCal $(k-1)$ times for each new predictor. However, running ReDCal sequentially does not minimize disagreement among all predictors. For instance, initially reconciling $f_1$ with $f_2$ minimizes their disagreement, but after $f_1$ undergoes further reconciliations with other predictors, its updated version $f_1^t$ may still end up disagreeing with $f_2$ for sufficiently many units. This disagreement indicates a falsification potential and suggests room for further improvements in $f_1$.

To address these challenges, we introduce a "contestation" method. We select a base model, $f$, and focus updates on events where $f$ disagrees with the other models, $f_j$. Using Definition 3.1, a disagreement event $E_{\ell,a_1,a_2}^\alpha(f(x), f_j(x), x)$ in this setting is identified by a pair of actions $(a_1, a_2)$, a loss function $\ell$, and a contestant predictor $f_j$. At each iteration, we select the disagreement event with the largest size and "patch" the model whose estimated loss on the event has a larger error. The detailed algorithm is written in Algorithm 4 and theoretical guarantees are in Theorem E.1.

To maintain the same guarantee as in Theorem 3.5, we need roughly $k$ times more samples for Algorithm 4 compared to ReDCal. Since each iteration decreases the Brier score of one predictor, the cumulative number of necessary updates across $k$ predictors, i.e., the total number of time-steps, is approximately multiplied by $O(k)$. The sample complexity then scales up only linearly with $k$. The detailed algorithm, formal theorem, and its proof are in Appendix E.

## 4 EXPERIMENTS

In this section, we complement our theoretical results with a set of experiments on real-world datasets to show our improvement in decreasing decision-making loss compared to prior work.

### 4.1 IMAGENET MULTI-CLASS CLASSIFICATION

**Experiment Setup.** We use the ImageNet dataset (Deng et al., 2009) and two pre-trained models provided by pyTorch (inception-v3 (Szegedy et al., 2015) and resnet50 (He et al., 2015)). Among the 50000 validation samples, we use 40000 samples for calibration and 10000 samples for testing.

We investigate how the downstream decision loss changes with the four calibration algorithms: Reconcile (Algorithm 3), Decision-Calibration (Algorithm 1), ReDCal (Algorithm 2), and the combination of running ReDCal after Decision Calibration as post-process. We run each calibration algorithm 500 times. For each run, we first randomly draw 100 classes from the 1000 classes of ImageNet. Then, we randomly generate a loss function such that, for each $y \in \mathcal{Y}$, $a \in \mathcal{A}$, $\ell(y, a) \sim \text{Normal}(0, 1)$. For each randomly generated loss function $\ell$, we compare the expected losses derived from the best-response policies based on predictors $f_1$ and $f_2$ against those based on the optimal predictor $f^*$. Formally, the loss gap at timestep $t$ using predictor $f_i$ is defined as

$$\text{LossGap}(f_i^t) = \mathbb{E}_{(x,y)\sim\mathcal{D}}[\ell(y, \pi_\ell^{\text{BR}}(f_i^t(x))) - \ell(y, \pi_\ell^{\text{BR}}(f^*(x)))].$$

The hyperparameters are chosen as follows: loss margin $\alpha = 0.001$, disagreement region mass $\eta = 0.01$, decision-calibration tolerance $\beta = 0.00001$, and the number of actions $K = 10$.

**Results.** The results are shown in Figure 2. Compared to Reconcile, ReDCal converges within a similar number of time-step and decreases the loss by a larger amount on the test dataset. Moreover, ReDCal further decrease the loss when used as a post-process after Decision-Calibration terminates.

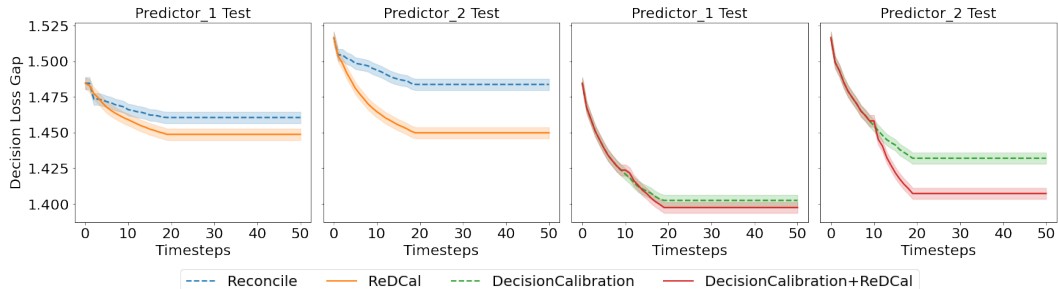

Figure 2: ReDCal decreases decision loss on Imagenet. We plot the gap between the average loss of each predictor and the optimal loss had we known the true predictor $f^*$ on the test set. **Left two figures:** We compare the LossGap of ReDCal (orange) with Reconcile (blue). Compared to Reconcile, our algorithm converges at a similar rate and decreases the loss by a larger amount on the test dataset. **Right two figures:** We compare the LossGap of Decision-Calibration (green) to that of Decision-Calibration with a run of ReDCal as post-process (red). Our proposed algorithm can improve upon Decision-Calibration output predictors and further reduce the loss on the test dataset. Results are averaged over $500$ runs and the shaded region indicates $\pm 1$ standard errors.

## 4.2 HAM10000 Multi-class Classification

We use the HAM10000 dataset (Tschandl et al., 2018) on pigmented skin lesions to predict the probability that a patient has contracted one of 7 possible skin diseases. The results are shown in Figure 3. For a detailed description and discussion of the results of this experiment, see Appendix G.

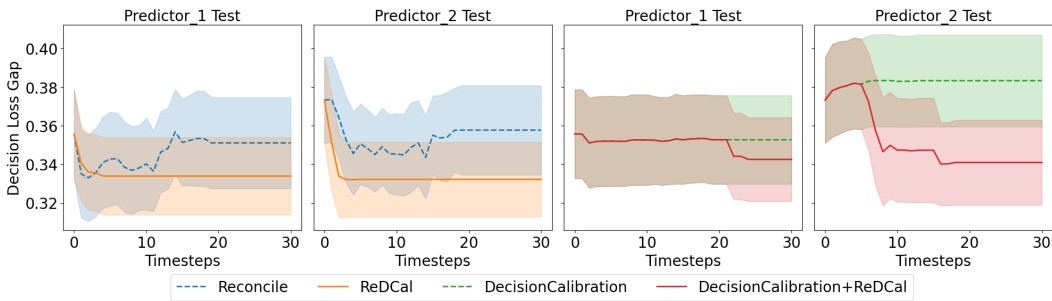

Figure 3: ReDCal decreases decision loss on the HAM10000 dataset. The comparisons and results are similar to that of Figure 2. Results are averaged over $10$ runs and the shaded region indicates $\pm 1$ standard errors.

## 5 Conclusion

Predictive multiplicity is a phenomenon in machine learning where the decision-makers have multiple predictors with nearly equivalent squared loss but vastly different individual predictions. Leveraging technique from the multi-calibration literature, we propose an algorithm, ReDCal, that updates a pair of predictors until they approximately agree almost everywhere on (1) individual predictions, (2) best-response actions in the downstream decision-making task, and (3) following the best-response actions incur losses that are close to the optimal loss. We further generalize this result to the settings where one has more than two predictive models and infinitely many possible downstream actions. Our theoretical results help alleviate the problem of predictive multiplicity in model selection. Finally, we provide experiments using real-world datasets to show that our proposed algorithm achieves lower decision loss compared to existing work. While we do not provide examples of domain-specific loss functions as part of our analysis and experiments, we hope that our findings can aid future studies on the impact of model multiplicity in decision-making.

ACKNOWLEDGMENTS

Z. S. W. and D. D. N were supported in part by an NSF grant # 1952085 and STTR grant.

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

# APPENDIX

## A  ADDITIONAL RELATED WORK

Within the literature on multigroup fairness, Kleinberg et al. (2016) highlighted the trade-offs in fairness by proving the incompatibility of three key fairness conditions—calibration, positive class balance, and negative class balance—emphasizing the need for prioritization in fairness criteria. Building on these foundational insights, Kim et al. (2019) introduced multiaccuracy. This post-processing technique ensures consistent accuracy across subpopulations without access to group labels, laying the baseline for future work on predictive fairness. Following such framework, Rothblum & Yona (2021) extended the PAC learning paradigm to multi-group scenarios. Shen et al.

(2023) proposed a framework for fair representations across multiple fairness notions and unknown tasks. More recently, Hu et al. (2024) explored robustness in multigroup fairness and proposed algorithms tailored for subpopulations under data corruption. Liu & Molinari (2024) proposed a method to estimate the fairness-accuracy frontier for testing fairness in decision-making. Another line of work (Haghtalab et al., 2023) connects fairness and optimization with a unified game theoretic framework. However, our work does not focus on the fairness aspect of multicalibration but instead borrows its techniques.

## B   LIMITATION OF PRIOR WORK (CONTINUE)

We provide the Algorithm 3 from (Roth et al., 2023) and their theoretical guarantees for completion. First, given two predictors $f_1$ and $f_2$, define the disagreement region as:

$$U_\epsilon(f_1, f_2) := \{x : |f_1(x) - f_2(x)| > \epsilon\}$$

which can be further divided into two partitions:

$$U_\epsilon^>(f_1, f_2) = \{x \in U_\epsilon(f_1, f_2) : f_1(x) > f_2(x)\}$$
$$U_\epsilon^<(f_1, f_2) = \{x \in U_\epsilon(f_1, f_2) : f_1(x) < f_2(x)\}$$

---

**Algorithm 3:** Reconcile (Roth et al., 2023)

---

**Input:** $f_1, f_2, \eta > 0, \alpha > 0$

1: Let $f_1^0 = f_1, f_2^0 = f_2$.

2: **while** $\mu(U_\alpha(f_1^{t_1}, f_2^{t_2})) \geq \eta$ **do**

3:   For each $\bullet \in \{>, <\}$ and $i \in \{1, 2\}$, let:

$$v_*^\bullet = \mathbb{E}[y | x \in U_\epsilon^\bullet(f_1^{t_1}, f_2)] \quad v_i^\bullet = \mathbb{E}[f_i^{t_i}(x) | x \in U_\epsilon^\bullet(f_1^{t_1}, f_2)]$$

4:   Let

$$(i_t, \bullet_t) = \underset{i \in \{1,2\}, \bullet \in \{>, <\}}{\arg\max} \, \mu(U_\epsilon^\bullet(f_1^{t_1}, f_2^{t_2})) \cdot (v_*^\bullet - v_i^\bullet)^2$$

  breaking ties arbitrarily.

5:   Let:

$$g_t(x) = \begin{cases} 1 & x \in U_\epsilon^{\bullet_t}(f_1^{t_1}, f_2^{t_2}) \\ 0 & \text{otherwise} \end{cases}$$

6:   Let

$$\widetilde{\Delta}_t = \mathbb{E}_{(x,y) \sim \mathcal{D}}[y | g_t(x) = 1] - \mathbb{E}_{(x,y) \sim \mathcal{D}}[f_{i_t}^{t_{i_t}}(x) | g_t(x) = 1]$$
$$\Delta_t = \text{Round}(\widetilde{\Delta}_t; m)$$

7:   Let $f_i^{t_i+1}(x) = \text{proj}_{[0,1]^d}(f_i^{t_i}(x) + \Delta_t g_t(x)), t_i = t_i + 1, t = t + 1.$

**Output:** $(f_1^{t_1}, f_2^{t_2})$

---

**Theorem B.1** (Reconcile (Roth et al., 2023)). *For any pair of models $f_1, f_2 : \mathcal{X} \to [0, 1]$, any distribution $\mathcal{D}$, and any $\alpha, \eta > 0$, Algorithm 3 runs for $T = T_1 + T_2$ many rounds and outputs a pair of models $(f_1^{T_1}, f_2^{T_2})$ such that:*

1. $T \leq (B(f_1, \mathcal{D}) + B(f_2, \mathcal{D})) \cdot \frac{16}{\eta \alpha^2}$

2. $B(f_1^{T_1}, \mathcal{D}) \leq B(f_1, \mathcal{D}) - T_1 \cdot \frac{\eta \alpha^2}{16}$ and $B(f_2^{T_2}, \mathcal{D}) \leq B(f_2, \mathcal{D}) - T_2 \cdot \frac{\eta \alpha^2}{16}$

3. $\mu(U_\epsilon(f_1^{T_1}, f_2^{t_2})) \leq \eta$

Furthermore, we provide a counterexample that shows ensuring each individual predictor is decision-calibrated by running Algorithm 1 is not sufficient.

**Decision-Calibrated predictions.** Another baseline algorithm we consider is to run Algorithm 1 separately for both $f_1, f_2$. However, in the following theorem, we show that the updated predictors $f_1'$ and $f_2'$ can still disagree with each other on the best-response actions for substantially many units, indicating room for further improvement.

**Theorem B.2.** *For any $\eta \in (0, 1/4)$ and $\beta \in (0, 1/2)$, there exists a pair of predictors $f_1, f_2$ and a loss function $\ell$, such that after running Decision-Calibration (Zhao et al., 2021), the resulting models $f_1^T, f_2^T$ are $\beta$-decision-calibrated with respect to the loss function $\ell$. There exists a set of units $x$ with probability mass $2\eta$ where $f_1^T$ and $f_2^T$ disagree on the individual best-response actions.*

*Proof.* For any $\eta \in (0, 1/4), \beta \in (0, 1/2)$, let $\mathcal{X} = [4]$, with $\Pr[1] = \Pr[4] = 1/2 - \eta$ and $\Pr[x = 2] = \Pr[x = 3] = \eta$. Consider the predictors $f_1, f_2$ as follows:

$$f_1(1) = f_1(2) = f_2(1) = f_2(3) = \frac{\beta}{4} - 2\eta\beta + 2\eta, \tag{5}$$

$$f_1(3) = f_1(4) = f_2(2) = f_2(4) = 1 - \frac{\beta}{2}, \tag{6}$$

$$f^*(1) = \frac{\beta}{2}, \quad f^*(2) = f^*(3) = f^*(4) = 1 - \frac{\beta}{2}. \tag{7}$$

Notice that

$$\frac{\beta}{4} - 2\eta\beta + 2\eta = \frac{\beta}{4} + 2\eta(1 - \beta) < \frac{1}{4} + \frac{1}{4} = \frac{1}{2}.$$

The best-response policy for each predictor is

$$\pi_\ell^{\mathsf{BR}}(f_1(1)) = \pi_\ell^{\mathsf{BR}}(f_1(2)) = 0, \quad \pi_\ell^{\mathsf{BR}}(f_1(3)) = \pi_\ell^{\mathsf{BR}}(f_1(4)) = 1, \tag{8}$$

$$\pi_\ell^{\mathsf{BR}}(f_2(1)) = \pi_\ell^{\mathsf{BR}}(f_2(3)) = 0, \quad \pi_\ell^{\mathsf{BR}}(f_2(2)) = \pi_\ell^{\mathsf{BR}}(f_1(4)) = 1. \tag{9}$$

For each best-response event, we have

$$\mathbb{E}_{(x,y)\sim\mathcal{D}}[f^*(x)E_1(f_1(x), x)] = \mathbb{E}_{(x,y)\sim\mathcal{D}}[f_1(x)E_1(f_1(x), x)],$$
$$\mathbb{E}_{(x,y)\sim\mathcal{D}}[f^*(x)E_2(f_1(x), x)] = \mathbb{E}_{(x,y)\sim\mathcal{D}}[f_1(x)E_2(f_1(x), x)],$$
$$\mathbb{E}_{(x,y)\sim\mathcal{D}}[f^*(x)E_1(f_2(x), x)] = \mathbb{E}_{(x,y)\sim\mathcal{D}}[f_2(x)E_1(f_2(x), x)],$$
$$\mathbb{E}_{(x,y)\sim\mathcal{D}}[f^*(x)E_2(f_2(x), x)] = \mathbb{E}_{(x,y)\sim\mathcal{D}}[f_2(x)E_2(f_2(x), x)].$$

That is, $f_1, f_2$ are already decision-calibrated, so running Decision Calibration will not further improve either of the two predictors. However, based on our definition of disagreement events, we still have

$$E_{1,2} = \{2\}, \quad E_{2,1} = \{1\},$$

each with size

$$\mu(E_{1,2}) = \mu(E_{2,1}) = \eta.$$

However, we observe that $f_1$ and $f_2$ still disagree on the best-response action for units $x = 2$ and $x = 3$. We can further reduce the differences in best-response actions using our algorithm Algorithm 2. $\qquad \square$

## C  PROOFS OF SECTION 3.1: RECONCILE FOR DECISION MAKING

First, we show that if a disagreement event has a large probability mass, then at least one of $f_1, f_2$ has a large prediction error within the region:

**Lemma C.1.** *Fix any two predictors $f_1, f_2 : \mathcal{X} \to [0, 1]^d$ and $\alpha, \eta > 0$. If $\mu(E_{\ell, a_1, a_2}) > \eta$ for some $a_1, a_2 \in \mathcal{A}$, then we have*

$$\left\| \mathbb{E}_{x\sim\mathcal{X}}\left[ f_i(x) - f^*(x) \big| E_{\ell, a_1, a_2}(x) \right] \right\| \geq \frac{\alpha}{2\sqrt{d}} \tag{10}$$

*for some $i \in \{1, 2\}$.*

*Proof.* By definition of event $E_{\ell,a_1,a_2}$, we have

$$\mathbb{E}_{x\sim\mathcal{X}}\left[\ell(f_1(x),a_2) - \ell(f_1(x),a_1) + \ell(f_2(x),a_1) - \ell(f_2(x),a_2))|E_{\ell,a_1,a_2}(x)\right] \geq \alpha$$

Also, we have

$$
\begin{aligned}
&\mathbb{E}_{x\sim\mathcal{X}}\left[\ell(f_1(x),a_2) - \ell(f_1(x),a_1) + \ell(f_2(x),a_1) - \ell(f_2(x),a_2))|E_{\ell,a_1,a_2}(x)\right]\\
=&\mathbb{E}_{x\sim\mathcal{X}}\left[\langle f_1(x) - f_2(x), \ell_{a_2} - \ell_{a_1}\rangle|E_{\ell,a_1,a_2}(x)\right]\\
=&\mathbb{E}_{x\sim\mathcal{X}}\left[\langle f_1(x) - f^*(x), \ell_{a_2} - \ell_{a_1}\rangle|E_{\ell,a_1,a_2}(x)\right]\\
&+ \mathbb{E}_{x\sim\mathcal{X}}\left[\langle f^*(x) - f_2(x), \ell_{a_2} - \ell_{a_1}\rangle|E_{\ell,a_1,a_2}(x)\right]\\
\leq&\|\mathbb{E}_{x\sim\mathcal{X}}\left[f_1(x) - f^*(x)|E_{\ell,a_1,a_2}(x)\right]\|_2\sqrt{d} + \|\mathbb{E}_{x\sim\mathcal{X}}\left[f_2(x) - f^*(x)|E_{\ell,a_1,a_2}(x)\right]\|_2\sqrt{d}\\
=&\sqrt{d}\cdot\left(\|\mathbb{E}_{x\sim\mathcal{X}}\left[f_1(x) - f^*(x)|E_{\ell,a_1,a_2}(x)\right]\|_2 + \|\mathbb{E}_{x\sim\mathcal{X}}\left[f_2(x) - f^*(x)|E_{\ell,a_1,a_2}(x)\right]\|_2\right)
\end{aligned}
$$

where the last inequality comes from Cauchy-Schwartz and that $\ell$ is bounded in $[0,1]$.

Combining the above inequalities, we have

$$\sqrt{d}\cdot\left(\left\|\mathbb{E}_{x\sim\mathcal{X}}\left[f_1(x) - f^*(x)|E_{\ell,a_1,a_2}(x)\right]\right\|_2 + \left\|\mathbb{E}_{x\sim\mathcal{X}}\left[f_2(x) - f^*(x)|E_{\ell,a_1,a_2}(x)\right]\right\|_2\right) \geq \alpha.$$

Therefore, for some $i \in \{1,2\}$, we have

$$\left\|\mathbb{E}_{x\sim\mathcal{X}}\left[f_i(x) - f^*(x)|E_{\ell,a_1,a_2}(x)\right]\right\|_2 \geq \frac{\alpha}{2\sqrt{d}}.$$

$\square$

This lemma indicates that, if we have two predictors $f_1, f_2$ that create a large disagreement event, we can falsify at least one of the models. We now show that these events also provide a directly actionable way to improve one of the models.

**Lemma C.2.** *For any predictor* $f : \mathcal{X} \to [0,1]^d$, *any event* $E \in \mathcal{E}$, *and distribution* $\mathcal{D}$. *Let* $\phi = \mathbb{E}_{(x,y)\sim\mathcal{D}}[y - f(x)|E(x) = 1]$. *We patch $f$ as*

$$f'(x) = \text{proj}_{[0,1]^d}(f(x) + \phi E(x)), \quad where \quad \text{proj}_{[0,1]^d}(y) = \underset{y'\in[0,1]^d}{\arg\min}\|y - y'\|_2.$$

*Then,*

$$B(f,\mathcal{D}) - B(f',\mathcal{D}) \geq \|\phi\|_2^2\mu(E).$$

*Proof.*

$$
\begin{aligned}
B(f,\mathcal{D}) - B(f',\mathcal{D}) =&\mathbb{E}\left[\|f(x) - y\|_2^2 - \|f'(x) - y\|_2^2\right]\\
\geq&\mathbb{E}\left[\|f(x) - y\|_2^2 - \|f(x) + \phi E(x) - y\|_2^2\right]\\
&\qquad\qquad\text{(since projection is non-expansive)}\\
=&\mathbb{E}\left[2\langle y - f(x), \phi E(x)\rangle - \|\phi E(x)\|_2^2\right]\\
\geq&\|\phi\|_2^2\cdot\mu(E)
\end{aligned}
$$

$\square$

Therefore, whenever we have two predictors that have a large disagreement event, we can always falsify at least one of the predictors and improve it through patching, causing the Brier score to decrease by a large amount. Similarly, for a fixed predictor, if one of its best-response events has a large calibration error, we can patch the predictor within the event to decrease the Brier score. As the Brier score is bounded in $[0,d]$, these two observations imply that the number of time-steps for both Algorithm 2 and its subroutine Algorithm 1 are bounded.

Other than the Brier score, we also care about minimizing the loss of the downstream decision-making task. We now show that, after a further update through the subroutine Algorithm 1, the loss does not increase much at each time-step of Algorithm 2:

**Lemma C.3.** *For any predictors $f_1$, $f_2$, loss function $\ell \in \mathcal{L}$ and any distribution $\mathcal{D}$, at any time-step $t$ of Algorithm 2, the predictors satisfies*

$$\mathbb{E}_{(x,y)\sim\mathcal{D}}[\ell(y, \pi_\ell^{\mathsf{BR}}(f_i^{t+1}(x))) - \ell(y, \pi_\ell^{\mathsf{BR}}(f_i^t(x)))] \leq \beta\sqrt{d}A,$$

*for all $i \in \{1, 2\}$.*

*Proof.* At each round, we define the set $\Delta_a^t \subseteq E^t$ as

$$\Delta_a^t = \{x \in E^t : \pi_\ell^{\mathsf{BR}}(f_i^{t+1}(x)) = a\}.$$

Then, we have

$$\mathbb{E}_{(x,y)\sim\mathcal{D}}[\ell(y, \pi_\ell^{\mathsf{BR}}(f_i^{t+1}(x))) - \ell(y, \pi_\ell^{\mathsf{BR}}(f_i^t(x)))]$$
$$= \sum_{a\in\mathcal{A}} \mathbb{E}_{(x,y)\sim\mathcal{D}}[(\ell(y, a) - \ell(y, a_i^t))\Delta_a^t(x)].$$

For each term in the summation, we can upper-bound it as

$$\mathbb{E}_{(x,y)\sim\mathcal{D}}[(\ell(y, a) - \ell(y, a_i^t))\Delta_a^t(x)] \tag{11}$$
$$= \langle \mathbb{E}_{(x,y)\sim\mathcal{D}}[y\Delta_a^t(x)], \ell_a - \ell_{a_i^t} \rangle \qquad \text{(Linearity of Expectation)}$$
$$\leq \langle \mathbb{E}_{x\sim\mathcal{D}_\mathcal{X}}[f_i^{t+1}(x)\Delta_a^t(x)], \ell_a - \ell_{a_i^t} \rangle + \beta\sqrt{d} \qquad \text{(Since } f_i^{t+1} \text{ is } \beta\text{-calibrated)}$$
$$\leq \beta\sqrt{d}. \qquad \text{(Since } a \text{ is the new Best-response action)}$$

Summing these actions together, we have

$$\mathbb{E}_{(x,y)\sim\mathcal{D}}[\ell(y, \pi_\ell^{\mathsf{BR}}(f_i^{t+1}(x))) - \ell(y, \pi_\ell^{\mathsf{BR}}(f_i^t(x)))] \leq \beta\sqrt{d}A. \tag{12}$$

$\square$

Instead of setting a fixed $\beta$, we can calculate a different $\beta^t$ at each round, which allows a smaller increase in loss.

**Lemma C.4.** *At each round $t$, if $a_i^t$ is not the best action on $E^t$ in average, i.e.*

$$\delta^t = \max_{a\in\mathcal{A}} \mathbb{E}_{(x,y)\sim\mathcal{D}}[(\ell(y, a_i^t) - \ell(y, a))E^t(x)] > 0,$$

*then we can set $\beta^t \leq \delta^t/\sqrt{d}$, such that*

$$\mathbb{E}_{(x,y)\sim\mathcal{D}}[\ell(y, \pi_\ell^{\mathsf{BR}}(f_i^{t+1}(x))) - \ell(y, \pi_\ell^{\mathsf{BR}}(f_i^t(x)))] \leq 0.$$

*Proof.* We can write the change in loss at each round as

$$\mathbb{E}_{(x,y)\sim\mathcal{D}}[\ell(y, \pi_\ell^{\mathsf{BR}}(f_i^{t+1}(x))) - \ell(y, \pi_\ell^{\mathsf{BR}}(f_i^t(x)))]$$
$$= \mathbb{E}_{(x,y)\sim\mathcal{D}}[\ell(y, \pi_\ell^{\mathsf{BR}}(f_i^{t+1}(x))) - \ell(y, a_i^t))E^t(x)]$$
$$= \mathbb{E}_{(x,y)\sim\mathcal{D}}[\ell(y, \pi_\ell^{\mathsf{BR}}(f_i^{t+1}(x))) - \ell(y, a'))E^t(x)] + \mathbb{E}_{(x,y)\sim\mathcal{D}}[\ell(y, a') - \ell(y, a_i^t))E^t(x)]$$
$$= \sum_{a\in\mathcal{A}} \mathbb{E}_{(x,y)\sim\mathcal{D}}[(\ell(y, a) - \ell(y, a'))\Delta_a^t(x)] + \mathbb{E}_{(x,y)\sim\mathcal{D}}[\ell(y, a') - \ell(y, a_i^t))E^t(x)],$$

for any $a' \in \mathcal{A}$.

We can use the same analysis as in Lemma C.3 to get

$$\sum_{a\in\mathcal{A}} \mathbb{E}_{(x,y)\sim\mathcal{D}}[(\ell(y, a) - \ell(y, a'))\Delta_a^t(x)] \leq \beta^t\sqrt{d}.$$

For the second term, we would want the loss to be as small as possible, so we can choose $a' = \arg\min_{a\in\mathcal{A}} \mathbb{E}_{(x,y)\sim\mathcal{D}}[\ell(y, a) \cdot E^t(x)]$ and let

$$\delta^t = -\mathbb{E}_{(x,y)\sim\mathcal{D}}[(\ell(y, a') - \ell(y, a_i^t))E^t(x)], \tag{13}$$

then $\delta^t$ is maximized and $\delta^t \geq 0$ by definition.

The total change in loss in this round can be written as

$$\mathbb{E}_{(x,y)\sim\mathcal{D}}[\ell(y, \pi_\ell^{\mathsf{BR}}(f_i^{t+1}(x))) - \ell(y, \pi_\ell^{\mathsf{BR}}(f_i^t(x)))] \leq \beta^t \sqrt{d} - \delta^t.$$

If $\delta^t > 0$, we can set $\beta^t \leq \delta^t/\sqrt{d}$ to ensure the loss does not increase at this round.

$\square$

### C.1 Proof of Theorem 3.3

*Proof.* By Lemma C.1 and C.2, for any $i \in \{1, 2\}$, at time-step $t$, we have the inequality

$$B(f_i^t, \mathcal{D}) - B(f_i^{t+1}, \mathcal{D}) \geq \frac{\alpha^2 \eta}{4d}.$$

Taking the sum over all time-steps, we have for any $i \in \{1, 2\}$,

$$B(f_i, \mathcal{D}) - B(f_i^T, \mathcal{D}) \geq T_i \cdot \frac{\alpha^2 \eta}{4d}.$$

Since the Brier score is always non-negative, we have

$$T_i \leq \frac{4d \cdot B(f_i, \mathcal{D})}{\alpha^2 \eta}.$$

Second, using Lemma C.3 and summing over all time-steps, we have

$$\mathbb{E}_{(x,y)\sim\mathcal{D}}[\ell(y, \pi_\ell^{\mathsf{BR}}(f_i^T(x)))] - \mathbb{E}_{(x,y)\sim\mathcal{D}}[\ell(y, \pi_\ell^{\mathsf{BR}}(f_i(x)))] \tag{14}$$

$$= \sum_{t=1}^T \mathbb{I}[i_t = i]\mathbb{E}_{(x,y)\sim\mathcal{D}}[\ell(y, \pi_\ell^{\mathsf{BR}}(f_i^{t+1}(x))) - \ell(y, \pi_\ell^{\mathsf{BR}}(f_i^t(x)))] \tag{15}$$

$$\leq T_i \cdot \beta\sqrt{d}A. \tag{16}$$

Finally, the halting condition implies that $\mu(E_{\ell,a_1,a_2}) < \eta$ for all $a_1, a_2 \in \mathcal{A}$. $\square$

## D Proofs of Section 3.2: Finite Sample Analysis

First, to make our argument that in-sample quantities translate to out-sample quantities, it is useful for the patching operations to use values that are rounded to a finite grid, rather than the precise value from the arbitrary sample. We define the finite grid as follows:

**Definition D.1.** *For any integer $m > 0$, let $1/m$ denote the $m + 1$ grid points,*

$$\left[\frac{1}{m}\right] = \left\{0, \frac{1}{m}, \frac{2}{m}, \ldots, \frac{m-1}{m}, 1\right\}.$$

*For any value $v \in [0,1]^d$, let $Round(v; m) = \arg\min_{v'\in[1/m]^d} \|v - v'\|_2$ denote the closest grid point to $v$ in $[1/m]^d$.*

At each time-step in Algorithm 1 and Algorithm 2, denote $\widetilde{\phi}^t = Round(\phi; m)$, and we patch the predictors using $\widetilde{\phi}^t$ instead of $\phi^t$, i.e., we update $f^t$ to $f^{t+1}$ as

$$f^{t+1}(x) = \text{proj}_{[0,1]^d}(f^t(x) + \widetilde{\phi}^t E^t(f^t(x), x)).$$

With this new patching operation, we can perform a similar analysis in Section 3.1 to show that the Brier score decreases at each iteration, and therefore the algorithm terminates within a finite number of time-steps. We denote the maximum number of time-steps, counting both Algorithm 1 and Algorithm 2, as $T_{\max}$.

Then, we can count the total number of possible predictors outputted by Algorithm 2 by observing that, for a fixed pair of input predictors, each pair of output predictors can be encoded as a sequence of tuple, $\{(i^t, E^t, \Delta^t)\}_{t \in [T]}$. Here, index $i^t \in \{1, 2\}$, event $E^t \in \{E_{\ell, a_1, a_2} : \ell \in \mathcal{L}, a_1, a_2 \in \mathcal{A}\} \cup \{E_{\ell, a} \cap E_{\ell', a_1, a_2} : \ell, \ell' \in \mathcal{L}, a_1, a_2, a \in \mathcal{A}\}$, and $\Delta_t \in [1/m]^d$ are all chosen from a finite set, and the length of the sequence, $T$, is also bounded. Specifically, for a fixed input $f_1, f_2$, we denote $S$ to be the set of all possible predictors outputted by Algorithm 2. Then, its size satisfies

$$|S| \le (4|\mathcal{L}|^2 A^3 (m+1)^d)^{T_{\max}+1}$$

We show that the number of predictors outputted by Algorithm 2 is bounded:

**Lemma D.2.** *Fix any pair of predictors $f_1, f_2 : \mathcal{X} \to [0, 1]^d$ and any $\eta, \alpha, \beta > 0$. Then the total number of possible predictors outputted by Algorithm 2 is at most $|S|$ such that, for any distribution $\mathcal{D}$ on which Algorithm 2 is run, the output predictors $(f_1^t, f_2^t) \in S$.*

*Proof.* First, notice that a sequence of quantities $\{(i^t, E^t, \Delta^t)\}_{t \in [T]}$ defines the pair of predictors outputted by Algorithm 2.

Let $S$ denote the pairs of functions induced by all such trajectories defined above. Here, $i^t \in \{1, 2\}, E^t \in \{E_{\ell, a_1, a_2} : \ell \in \mathcal{L}, a_1, a_2 \in \mathcal{A}\} \cup \{E_{\ell, a} \cap E_{\ell', a_1, a_2} : \ell, \ell' \in \mathcal{L}, a_1, a_2, a \in \mathcal{A}\}$, and $\Delta_t \in [1/m]^d$. Therefore, there are

$$|S| \le \sum_{t=1}^{T} \left( 2(|\mathcal{L}|A^2 + |\mathcal{L}|^2 A^3)(m+1)^d \right)^t \le (4|\mathcal{L}|^2 A^3 (m+1)^d)^{T_{\max}+1}$$

output predictors. $\qquad\qquad\square$

### D.1 FINITE GRID

With this new patching operation, we can show that the Brier score decreases on the empirical distribution $D$, corresponding to Lemma C.2:

**Lemma D.3.** *Fix any event $E$. Let $\phi = \mathbb{E}_{(x,y) \sim D}[y - f(x)|E(x) = 1]$. For any predictor $f$, we patch $f$ as $f'(x) = \operatorname{proj}_\Delta(f(x) + \tilde{\phi}E(x))$. Then,*

$$B(f, D) - B(f', D) \ge \|\phi\|_2^2 \mu(E) - \frac{d}{4m^2}$$

*Proof.* Let $\tilde{f}'(x) = f(x) + \phi E(x)$. Then, we have

$$
\begin{aligned}
B(f, D) - B(f', D) =& B(f, D) - B(\tilde{f}', D) + B(\tilde{f}', D) - B(f', D) \\
\ge& \|\phi\|_2^2 \mu(E) + \mathbb{E}[\|f(x) + \phi E(x) - y\|_2^2 - \left\|f(x) + \tilde{\phi}E(x) - y\right\|_2^2] \\
& \hspace{8cm} \text{(Lemma C.2)} \\
=& \|\phi\|_2^2 \mu(E) - \mathbb{E}\left[\left\|\tilde{\phi} - \phi\right\|_2^2\right] \mu(E)
\end{aligned}
$$

By definition $\tilde{\phi}$, we know that each index of $|\tilde{\phi} - \phi|$ is in $[0, \frac{1}{2m}]$. Therefore, we have

$$B(f, D) - B(f', D) \ge \|\phi\|_2^2 \cdot \mu(E) - \frac{d}{4m^2}.$$

$\qquad\qquad\square$

Since the Brier score is within the range $[0, d]$, and it decreases at each iteration, we can show that, if we set $m$ large enough, Algorithm 2 terminates within a finite number of iterations:

**Lemma D.4.** *For any predictor $f_1, f_2$. Let $m \ge \left\lceil \sqrt{\frac{d}{2 \min\{\beta^2, \eta\alpha^2/4d\}}} \right\rceil$. The Brier score at each iteration of Algorithm 1 and Algorithm 2 satisfies*

$$B(f^t, D) - B(f^{t+1}, D) > \frac{\min\{\beta^2, \eta\alpha^2/(4d)\}}{2}.$$

*Counting both Algorithm 2 and its subroutine Algorithm 1, the total number of iterations $T$ satisfies*

$$T \leq \frac{2d}{\min\{\beta^2, \eta\alpha^2/4d\}}.$$

*Proof.* By Algorithm 1, we have by definition of $\beta$-decision calibration that

$$\mu(E^t \cap E_{\ell,a}) \cdot \left\| \mathbb{E}_{(x,y)\sim D}[(y - f(x)|E^t(x) \cdot E_{\ell,a}(x) = 1]\right\|_2^2 \tag{17}$$

$$\geq \mu(E^t \cap E_{\ell,a})^2 \cdot \left\| \mathbb{E}_{(x,y)\sim D}[(y - f(x)|E^t(x) \cdot E_{\ell,a}(x) = 1]\right\|_2^2 \tag{18}$$

$$= \left\| \mathbb{E}_{(x,y)\sim D}[(y - f(x) \cdot E^t(x) \cdot E_{\ell,a}(x)]\right\|_2^2 > \beta^2. \tag{19}$$

In Algorithm 2, we have by Lemma C.1 that

$$\mu(E^t) \left\| \mathbb{E}_{(x,y)\sim D}[(y - f(x)|E^t(x) = 1]\right\|_2^2 > \frac{\eta\alpha^2}{4d}. \tag{20}$$

Therefore, for any $\phi$ and event $E$ that we patch in Algorithm 1 or Algorithm 2, they satisfy

$$\|\phi\|_2^2 \cdot \mu(E) > \min\{\beta^2, \frac{\eta\alpha^2}{4d}\}.$$

Letting $m \geq \left\lceil \sqrt{\frac{d}{2\min\{\beta^2, \eta\alpha^2/4d\}}} \right\rceil$, we can ensure

$$B(f^t, D) - B(f^{t+1}, D) \geq \frac{\|\phi\|_2^2 \cdot \mu(E)}{2} > \frac{\min\{\beta^2, \eta\alpha^2/(4d)\}}{2}.$$

Since the Brier score is in the range $[0, d]$, we can bound the total number of iterations of both Algorithm 1 and Algorithm 2 as

$$T \leq \frac{2d}{\min\{\beta^2, \eta\alpha^2/4d\}}.$$

$\square$

## D.2    PROOF OF THEOREM 3.5

First, we show that, for a fixed predictor $f$ and event $E_{\ell,a}$, the in-sample prediction error is approximately accurate. The deviation bound of the Brier score and calibration error can then be directly implied.

**Lemma D.5.** *Fix any $f$, $E_{\ell,a}$, with probability at least $1 - \delta'$, we have*

$$\left\| \mathbb{E}_{\mathcal{D}}[(y - f(x))E_{\ell,a}(f(x), x)] - \frac{1}{n}\sum_{i=1}^{n}[(y_i - f(x_i))E_{\ell,a}(f(x_i), x_i)] \right\|_2 \leq \sqrt{\frac{d\ln(2d/\delta')}{2n}}.$$

*Proof.* Fix an index $j \in [d]$, we know $\mathbb{E}_{(x,y)\sim\mathcal{D}}[(y - f(x))_j \cdot E_{\ell,a}(f(x), x)] \in [0, 1]$ and

$$\mathbb{E}_D\left[\frac{1}{n}\sum_{i=1}^{n}[(y_i - f(x_i))_j \cdot E_{\ell,a}(f(x_i), x_i)]\right] = \mathbb{E}_{(x,y)\sim\mathcal{D}}[(y - f(x))_j \cdot E_{\ell,a}(f(x), x)].$$

Since $(x_i, y_i)$ is drawn i.i.d. from $\mathcal{D}$, we can use Hoeffding's inequality to get, with probability $\delta'/d$,

$$\left| \mathbb{E}_{(x,y)\sim\mathcal{D}}[(y - f(x))_j \cdot E_{\ell,a}(f(x), x)] - \frac{1}{n}\sum_{i=1}^{n}[(y_i - f(x_i))_j \cdot E_{\ell,a}(f(x_i), x_i)] \right| \leq \sqrt{\frac{\ln(2d/\delta')}{2n}}$$

Using union bound, we have that with probability $1 - \delta'$, the above inequality holds for all $j \in [d]$. Then, we have

$$\left\| \mathbb{E}_{(x,y)\sim\mathcal{D}}[(y - f(x)) \cdot E_{\ell,a}(f(x), x)] - \frac{1}{n}\sum_{i=1}^{n}[(y_i - f(x_i)) \cdot E_{\ell,a}(f(x_i), x_i)] \right\|_2$$

$$\leq \sqrt{\sum_{j\in d}\left(\sqrt{\frac{\ln(2d/\delta')}{2n}}\right)^2} = \sqrt{\frac{d\ln(2d/\delta')}{2n}}.$$

$\square$

The deviation bound of the Brier score and calibration error can be directly implied by Lemma D.5. We summarize them in the lemmas below:

**Lemma D.6.** *For a fixed $f$, with probability at least $1 - \delta'$, $|B(f, \mathcal{D}) - B(f, D)| \leq \sqrt{\frac{d \ln(2d/\delta')}{2n}}$.*

*Proof.* Using triangle inequality, we have

$$
|B(f, \mathcal{D}) - B(f, D)| = \left| \left\| \mathbb{E}_{(x,y) \sim \mathcal{D}}[y - f(x)] \right\|_2 - \left\| \frac{1}{n} \sum_{i=1}^n [y_i - f(x_i)] \right\|_2 \right|
$$

$$
\leq \left\| \mathbb{E}_{(x,y) \sim \mathcal{D}}[y - f(x)] - \frac{1}{n} \sum_{i=1}^n [y_i - f(x_i)] \right\|_2
$$

$$
\leq \sqrt{\frac{d \ln(2d/\delta')}{2n}}.
$$

$\square$

**Lemma D.7.** *For a fixed $f$, any loss function $\ell \in \mathcal{L}$ and $\mathcal{E} = \{E_{\ell,a} \cap E_{\ell',a_1,a_2} : \ell, \ell' \in \mathcal{L}, a, a_1, a_2 \in \mathcal{A}\}$, with probability $1 - \delta'$, we have*

$$
\left\| \mathbb{E}_{\mathcal{D}}[(y - f(x))E(x)] - \frac{1}{n} \sum_{i=1}^n [(y_i - f(x_i))E(x)] \right\|_2 \leq \sqrt{\frac{3d \ln(2dA |\mathcal{L}| /\delta')}{2n}}
$$

*for all $E \in \mathcal{E}$.*

*Proof.* The claim follows by using a union bound over the events in $\mathcal{E}^t$, using Lemma D.5, and that $|\mathcal{E}^t| = A^3 |\mathcal{L}|^2$. $\square$

For a fixed pair of predictors, we can also show that the empirical size of the disagreement events $E_{\ell,a_1,a_2}$ is approximately correct with high probability:

**Lemma D.8.** *Fix any pair of predictors $(f_1, f_2) \in S$, with probability at least $1 - \delta'$ over $D$, we have*

$$
\left| \mu(E_{\ell,a_1,a_2}) - \frac{1}{n} \sum_{i=1}^n \mathbb{I}[E_{\ell,a_1,a_2}(x_i) = 1] \right| \leq \sqrt{\frac{2 \ln(2A |\mathcal{L}| /\delta')}{2n}}.
$$

*for all $a_1, a_2 \in \mathcal{A}$ with $a_1 \neq a_2$ and for all $\ell \in \mathcal{L}$.*

*Proof.* We know $\mathbb{I}[E_{\ell,a_1,a_2}(x_i) = 1] \in [0, 1]$ and

$$
\mathbb{E}_D \left[ \frac{1}{n} \sum_{i=1}^n \mathbb{I}[E_{\ell,a_1,a_2}(x_i)] \right] = \mu(E_{\ell,a_1,a_2}).
$$

Since $(x_i, y_i)$ is drawn i.i.d. from $\mathcal{D}$, we can use Hoeffding's inequality to get, with probability $1 - \delta'/(A^2|\mathcal{L}|)$,

$$
\left| \mu(E_{\ell,a_1,a_2}) - \frac{1}{n} \sum_{i=1}^n \mathbb{I}[E_{\ell,a_1,a_2}(x_i)] \right| \leq \sqrt{\frac{2 \ln(2A |\mathcal{L}| /\delta')}{2n}}.
$$

Using union bound over all pairs of $a_1, a_2 \in \mathcal{A}$ and $\ell \in \mathcal{L}$, we know the above inequality holds for all $a_1, a_2$ and $\ell \in \mathcal{L}$ with probability at least $1 - \delta'$. $\square$

We summarize the above results in the theorem below. Theorem 3.5 follows by solving for $n$ in the 2-4th guarantees below.

**Theorem D.9.** *Fix any distribution $\mathcal{D}$ and dataset $D \sim \mathcal{D}$ containing $n$ samples drawn i.i.d from $\mathcal{D}$. For any pair of predictors $f_1, f_2 : \mathcal{X} \to [0,1]^d$, loss margin $\alpha > 0$, disagreement region mass $\eta > 0$, and decision-calibration tolerance $\beta > 0$, Algorithm 2 run over the empirical distribution $D$ updates predictors $f_1$ and $f_2$ for $T_1$ and $T_2$ time-steps, respectively, and outputs a pair of predictors $(f_1^T, f_2^T)$ such that, with probability at least $1 - \delta$ over the randomness of $D \sim \mathcal{D}^n$,*

1. *The total number of time-steps for Algorithm 2 and Algorithm 1 is*

$$T = T_1 + T_2 \leq \frac{2d}{\min\{\beta^2, \eta\alpha^2/4d\}}$$

2. *For $i \in \{1, 2\}$, the Brier scores of the final models are lower than that of the input models:*

$$B(f_i^{T_i}, \mathcal{D}) \leq B(f_i, \mathcal{D}) - T_i \cdot \min\left\{\beta^2, \eta\alpha^2/(4d)\right\} + \sqrt{(d\ln(6d|S|/\delta)/(2n)}$$

3. *For $i \in \{1, 2\}$, the downstream decision-making losses of the final models do not increase by much compared to that of the input models:*

$$\mathbb{E}_{(x,y)\sim\mathcal{D}}[\ell(y, \pi^{\mathsf{BR}}(f_i^T(x)) - \ell(y, \pi^{\mathsf{BR}}(f_i(x)))] \leq \left(\beta + \sqrt{(3d\ln(6dA|S||\mathcal{L}|/\delta)/(2n)}\right)\sqrt{d}AT_i$$

4. *The final models approximately agree on their best-response actions almost everywhere. That is, the disagreement region $E_{a_1, a_2}$ calculated using $f_1^T, f_2^T$ has small mass.*

$$\mu(E_{\ell, a_1, a_2}) \leq \eta + \sqrt{(2\ln(6A|S||\mathcal{L}|/\delta)/(2n)} \quad \textit{for all } a_1, a_2 \in \mathcal{A} \quad \textit{s.t } a_1 \neq a_2$$

*Here, $S$ is the set of all possible predictors outputted by Algorithm 2 satisfying*

$$\ln(|S|) \leq \left(\frac{2d}{\min\{\beta^2, \eta\alpha^2/4d\}} + 1\right) \ln\left(4|\mathcal{L}|^2 A^3 \left(\left\lceil\sqrt{\frac{d}{2\min\{\beta^2, \eta\alpha^2/4d\}}}\right\rceil + 1\right)^d\right)$$

*Proof.* The upper bound on $T$ holds true with probability 1. For the remaining three guarantees, we show that each of them holds with probability at least $1 - \delta/3$ over the randomness of $D$.

**Brier Score**. First, by Lemma D.6 and using union bound over all possible output predictors $(f_1, f_2) \in S$, we have with probability at least $1 - \delta/3$ that

$$|B(f_i, \mathcal{D}) - B(f_i, D)| \leq \sqrt{\frac{d\ln(6d|S|/\delta)}{2n}}.$$

By Lemma D.4, and summing over all iterations, we have

$$B(f_i^{T_i}, D) \leq B(f_i, D) - T_i \cdot \min\left\{\frac{\beta^2}{2}, \frac{\eta\alpha^2}{8d}\right\}$$

Therefore,

$$B(f_i^{T_i}, \mathcal{D}) \leq B(f_i^{T_i}, D) + \sqrt{\frac{d\ln(6d|S|/\delta)}{2n}}$$

$$\leq B(f_i, \mathcal{D}) - T_i \cdot \min\left\{\frac{\beta^2}{2}, \frac{\eta\alpha^2}{8d}\right\} + \sqrt{\frac{d\ln(6d|S|/\delta)}{2n}}$$

for $i \in \{1, 2\}$.

**Expected Loss**. Using union bound over all predictors in $S$, by Lemma D.7, we have, with probability at least $1 - \delta/3$,

$$\left\|\mathbb{E}_{\mathcal{D}}[(y - f(x))E_{\ell,a}(f(x), x)] - \frac{1}{n}\sum_{i=1}^{n}[(y_i - f(x_i))E_{\ell,a}(f(x_i), x_i)]\right\|_2 \leq \sqrt{\frac{3d\ln(6dA|S||\mathcal{L}|/\delta)}{2n}}$$

for all predictors $f$, action $a \in \mathcal{A}$ and loss $\ell \in \mathcal{L}$. Using similar method as in Lemma 2.6, we define the set $\Delta_a^t \subseteq E^t$ as

$$\Delta_a^t = \{x \in E^t : \pi_\ell^{\mathsf{BR}}(f_i^{t+1}(x)) = a\}.$$

Then, we have

$$\mathbb{E}_{(x,y)\sim\mathcal{D}}[\ell(y,\pi_\ell^{\mathsf{BR}}(f_i^{t+1}(x)))-\ell(y,\pi_\ell^{\mathsf{BR}}(f_i^t(x)))]$$
$$=\sum_{a\in\mathcal{A}}\mathbb{E}_{(x,y)\sim\mathcal{D}}[(\ell(y,a)-\ell(y,a_i^t))\Delta_a^t(x)].$$

For each term in the summation,

$$\mathbb{E}_{(x,y)\sim\mathcal{D}}[(\ell(y,a)-\ell(y,a_i^t))\Delta_a^t(x)] \tag{21}$$
$$=\langle\mathbb{E}_{(x,y)\sim\mathcal{D}}[y\Delta_a^t(x)],\ell_a-\ell_{a_i^t}\rangle \qquad\text{(Linearity of Expectation)}$$
$$\leq\langle\mathbb{E}_{x\sim\mathcal{D}_\mathcal{X}}[f_i^{t+1}(x)\Delta_a^t(x)],\ell_a-\ell_{a_i^t}\rangle+\left(\beta+\sqrt{\frac{3d\ln(6dA|S|\,|\mathcal{L}|\,/\delta)}{2n}}\right)\sqrt{d}$$
$$\qquad\qquad\qquad\qquad\qquad\qquad\qquad\qquad\text{(Lemma D.7 and }\beta\text{-calibrated)}$$
$$\leq\left(\beta+\sqrt{\frac{3d\ln(6dA|S|\,|\mathcal{L}|\,/\delta)}{2n}}\right)\sqrt{d}. \qquad\text{(Since }a\text{ is the new Best-response action)}$$

Summing these actions together, we have

$$\mathbb{E}_{(x,y)\sim\mathcal{D}}[\ell(y,\pi_\ell^{\mathsf{BR}}(f_i^{t+1}(x)))-\ell(y,\pi_\ell^{\mathsf{BR}}(f_i^t(x)))]\leq\beta\sqrt{d}A+\sqrt{\frac{3\ln(6dA|S|\,|\mathcal{L}|\,/\delta)}{2n}}dA. \tag{22}$$

Summing over all iterations, we conclude that, with probability at least $1-\delta/3$,

$$\mathbb{E}_{(x,y)\sim\mathcal{D}}[\ell(y,\pi^{\mathsf{BR}}(f_i^{T_i}(x)))-\ell(y,\pi^{\mathsf{BR}}(f_i(x)))]\leq T_i(\beta\sqrt{d}A+\sqrt{\frac{3\ln(6dA|S|\,|\mathcal{L}|\,/\delta)}{2n}}dA)$$

for all $i\in\{1,2\}$.

**Disagreement Event**. By Lemma D.8, with probability at least $1-\delta/(3|S|)$, we have, for all $\ell\in\mathcal{L}$, $a_1,a_2\in\mathcal{A}$ with $a_1\neq a_2$ and all $(f_1,f_2)\in S$,

$$\left|\mu(E_{\ell,a_1,a_2}(f_1(x),f_2(x),x)-\frac{1}{n}\sum_{i=1}^n\mathbb{I}[E_{\ell,a_1,a_2}(f_1(x_i),f_2(x_i),x_i))]\right|\leq\sqrt{\frac{2\ln(6A|S|\,|\mathcal{L}|\,/\delta)}{2n}}.$$

From the while loop condition in Algorithm 2, we know that

$$\frac{1}{n}\sum_{i=1}^n\mathbb{I}[E_{\ell,a_1,a_2}(f_1^{T_1}(x_i),f_2^{T_2}(x_i),x_i)]\leq\eta$$

Then, using union bound over all $(f_1,f_2)\in S$ and $\ell\in\mathcal{L}$, with probability at least $1-\delta/3$, we have the guarantee

$$\mu(E_{\ell,a_1,a_2}(f_1^{T_1}(x),f_2^{T_2}(x),x))\leq\frac{1}{n}\sum_{i=1}^n\mathbb{I}[E_{\ell,a_1,a_2}(f_1^{T_1}(x_i),f_2^{T_2}(x_i),x_i)]+\sqrt{\frac{2\ln(6A|S|\,|\mathcal{L}|\,/\delta)}{2n}}$$
$$\tag{23}$$
$$\leq\eta+\sqrt{\frac{2\ln(6A|S|\,|\mathcal{L}|\,/\delta)}{2n}} \tag{24}$$

for all $a_1,a_2\in\mathcal{A}, a_1\neq a_2$ and $\ell\in\mathcal{L}$.

Finally, using results from Lemma D.2, value of $T_{\max}$, and $m=\left\lceil\sqrt{\frac{d}{2\min\{\beta^2,\eta\alpha^2/4d\}}}\right\rceil$, we conclude by showing

$$\ln(|S|)\leq\ln\left((4|\mathcal{L}|^2A^3(m+1)^d)^{T_{\max}+1}\right)$$
$$=(T_{\max}+1)\ln(4|\mathcal{L}|^2A^3(m+1)^d)$$
$$=\left(\frac{2d}{\min\{\beta^2,\eta\alpha^2/4d\}}+1\right)\ln\left(4|\mathcal{L}|^2A^3\left(\left\lceil\sqrt{\frac{d}{2\min\{\beta^2,\eta\alpha^2/4d\}}}\right\rceil+1\right)^d\right)$$

$$\square$$

# E PROOFS OF SECTION 3.3: EXTENSION TO MULTIPLE PREDICTORS

Here, we present the algorithm we described in Section 3.3. As a shorthand, we write $E_{\ell,a_1,a_2}^{\alpha}(f(x), f_j(x), x)$ as $E_{\ell,a_1,a_2,j}$.

---

**Algorithm 4:** Reconcile Decision Calibration for Multiple Predictors (ReDCal-Multi)

**Input:** $f_1, f_2, \ldots, f_k, \mathcal{L}, \eta > 0, \alpha > 0, \beta > 0$
1: Let $f_i^0 = f_i$ for all $i \in [k]\backslash\{1\}$, and $t = 0$. We use $f_1$ as the base predictor.
2: **while** $\mu(E_{\ell,a_1,a_2,j}) \geq \eta$ for some $a_1, a_2 \in \mathcal{A}, \ell \in \mathcal{L}$, and $j \in [k]\backslash\{1\}$ **do**
3:    Let $\ell^t, a_1^t, a_2^t, j^t = \arg\max_{\ell,a,a'} \mu(E_{\ell,a,a'}), E^t = E_{\ell^t, a_1^t, a_2^t, j^t}$.
4:    Pick

$$i^t = \arg\max_{i \in \{1,j^t\}} \left| \mathbb{E}_{(x,y)\sim\mathcal{D}}[\ell^t(y, a_1^t) - \ell^t(y, a_2^t)|E^t(x) = 1] \right.$$
$$\left. - \mathbb{E}_{x\sim\mathcal{X}}[\ell^t(f_i(x), a_1^t) - \ell^t(f_i(x), a_2^t)|E^t(x) = 1] \right|.$$

5:    Denote $f_{i^t}^t$ as $f_i^t$. Let $\phi^t = \mathbb{E}_{(x,y)\sim\mathcal{D}}[y|E^t(x) = 1] - \mathbb{E}_{x\sim\mathcal{D}_{\mathcal{X}}}[f_i^t(x)|E^t(x) = 1]$.
6:    Patch $f^t(x) = \mathrm{proj}_{[0,1]^d}(f_i^t(x) + \phi^t E^t(x))$.
7:    Let $f_i^{t+1} = \text{Decision-Calibration}(f^t, \mathcal{L}, \beta, E^t). t = t + 1$.
**Output:** $f_1^t, \ldots, f_k^t$

---

With Algorithm 4, we can obtain the following guarantee.

**Theorem E.1.** *Fix any distribution $\mathcal{D}$ and dataset $D \sim \mathcal{D}$ containing $n$ samples drawn i.i.d from $\mathcal{D}$. For any $k$ predictors $f_1, f_2, \ldots, f_k : \mathcal{X} \to [0,1]^d$, family of loss functions $\mathcal{L}$, loss margin $\alpha > 0$, disagreement region mass $\eta > 0$, and decision-calibration tolerance $\beta > 0$, Algorithm 4 run over the empirical distribution $D$ updates each $f_i$ for $T_i$ time-steps, and outputs $k$ predictors $(f_1^T, \ldots, f_k^T)$ such that, with probability at least $1 - \delta$ over the randomness of $D \sim \mathcal{D}^n$,*

1. *The total number of time-steps for Algorithm 4 and Algorithm 1 is*

$$T = \sum_{i \in [k]} T_i \leq \frac{2dk}{\min\{\beta^2, \eta\alpha^2/4d\}}$$

2. *For $i \in [k]$, the Brier scores of the final models are lower than that of the input models:*

$$B(f_i^{T_i}, \mathcal{D}) \leq B(f_i, \mathcal{D}) - (T_i - \eta) \cdot \min\left\{\beta^2, \eta\alpha^2/(4d)\right\}$$

3. *For $i \in [k]$ and for all $\ell \in \mathcal{L}$, the downstream decision-making losses of the final models do not increase by much compared to that of the input models:*

$$\mathbb{E}_{(x,y)\sim\mathcal{D}}[\ell(y, \pi_\ell^{\mathsf{BR}}(f_i^T(x)) - \ell(y, \pi_\ell^{\mathsf{BR}}(f_i(x))] \leq 2T_i\beta A\sqrt{d}$$

4. *The final models approximately agree on their best-response actions almost everywhere. That is, the disagreement region $E_{\ell,a_1,a_2,j}$ calculated using $f_1^T, f_j^T$ has small mass: For all $j \in [k], \ell \in \mathcal{L}$, and $a_1, a_2 \in \mathcal{A}$ s.t. $a_1 \neq a_2$,*

$$\mu(E_{\ell,a_1,a_2,j}) \leq 2\eta$$

*if $n \geq \Omega\left(d^2k/(\eta^2\min\{\beta, \eta\alpha^2/d\}) \cdot (\ln(dkA|\mathcal{L}|) - \ln(\delta\min\{\beta, \eta\alpha/d\}))\right)$.*

*Proof.* The proof follows the same idea and structure as the proof of theorem 3.5.

**Number of Iterations.** By lemma D.3, at each iteration of Algorithm 4 and Algorithm 1, the Brier score of some predictor $j \in [k]$ satisfies

$$B(f_j^t, D) - B(f_j^{t+1}, D) > \frac{\min\{\beta^2, \eta\alpha^2/(4d)\}}{2}.$$

Since the Brier score is non-negative and we have $k$ predictors in total, we have that, after $T = \sum_{i=1}^{k} T_i$ iterations,

$$\sum_{j \in [k]} B(f_j, D) \geq \sum_{j \in [k]} (B(f_j, D) - B(f_j^T, D)) > T \cdot \frac{\min\{\beta^2, \eta\alpha^2/(4d)\}}{2}.$$

Since Brier score is in the range $[0, d]$, we can solve for $T$ in the inequality and have the upper bound on the number of iterations

$$T_{\max} \leq \frac{2dk}{\min\{\beta^2, \eta\alpha^2/(4d)\}}.$$

**Brier Score.** We first count the number of possible predictors outputted by Algorithm 4. At each iteration in Algorithm 4 or its subroutine Algorithm 1, we patch on a predictor $f_i^t$ for some $i^t \in [k]$, on the event $E \in \{E_{\ell,a_1,a_2,j} : \ell \in \mathcal{L}, a_1, a_2 \in \mathcal{A}, j \in [k]\} \cup \{E_{\ell,a_1,a_2,j} \cap E_{\ell',a} : \ell, \ell' \in \mathcal{L}, a_1, a_2, a \in \mathcal{A}, j \in [k]\}$, with the value $\Delta_t \in [1/m]^d$. Therefore, at each iteration, the number of different patching operations satisfies

$$(|\mathcal{L}|A^2 k + |\mathcal{L}|^2 A^3 k)(m + 1)^d \leq 2|\mathcal{L}|^2 A^3 k(m + 1)^d.$$

For fixed $k$ predictors, the output predictors are defined by the sequence of patches. The total number of different sets of output predictors are

$$|S| \leq \sum_{i=1}^{T} (2|\mathcal{L}|^2 A^3 k(m + 1)^d)^t \leq (2|\mathcal{L}|^2 A^3 k(m + 1)^d)^{T_{\max}+1}.$$

Using lemma D.6 and using union bound over all possible output predictors, we have with probability at least $1 - \delta/3$ that

$$|B(f_i, D) - B(f_i, \mathcal{D})| \leq \sqrt{\frac{d \ln(6dk|S|/\delta)}{2n}}.$$

Summing over all iterations, we have

$$B(f_i^{T_i}, \mathcal{D}) \leq B(f_i^{T_i}, \mathcal{D}) - T_i \cdot \min\left\{\frac{\beta^2}{2}, \frac{\eta\alpha^2}{8d}\right\} + \sqrt{\frac{d \ln(6dk|S|/\delta)}{2n}}.$$

**Expected Loss.** Using lemma D.6 and a union bound over the events in $\mathcal{E} = \{E_{\ell,a_1,a_2,j} \cap E_{\ell',a} : \ell, \ell' \in \mathcal{L}, a_1, a_2, a \in \mathcal{A}, j \in [k]\}$, we have with probability $1 - \delta/3$ that

$$\left\|\mathbb{E}_{\mathcal{D}}[(y - f(x)]E(x) - \frac{1}{n}\sum_{i=1}^{n}[(y_i - f(x_i))E(x)]\right\| \leq \sqrt{\frac{3d \ln(6dkA|\mathcal{L}||S|/\delta)}{2n}}$$

for all predictors $f$, action $a \in \mathcal{A}$, loss $\ell \in \mathcal{L}$. The rest of the steps are the same as in the proof of theorem 3.5. We have, at iteration $t$,

$$\mathbb{E}_{(x,y)\sim\mathcal{D}}[\ell(y, \pi_\ell^{\mathsf{BR}}(f_i^{t+1}(x))) - \ell(y, \pi_\ell^{\mathsf{BR}}(f_i^t(x)))] \leq \beta\sqrt{d}A + \sqrt{\frac{3 \ln(6dkA|\mathcal{L}||S|/\delta)}{2n}}dA.$$

The result follows by summing over all iterations.

**Disagreement event is small.** The stopping condition at line 2 of algorithm 4 implies that the final output models satisfies

$$\mu(E_{\ell,a_1,a_2,j}) = \frac{1}{n}\sum_{i=1}^{n} E_{\ell,a_1,a_2,j}^\alpha(f_1^T(x_i), f_j^T(x_i), x_i) \leq \eta$$

for all $a_1, a_2 \in \mathcal{A}$, $\ell \in \mathcal{L}$, and $j \in [k]\backslash\{1\}$. That is, the disagreement region between our base model $f_1^T$ and any of the other $k - 1$ models is small. Using Hoeffding's inequality and union bound, we have that, for any predictors $(f_1, \ldots, f_k) \in S$, with probability $1 - \delta/3$ over $D$,

$$\left|\mu(E_{\ell,a_1,a_2,j} - \frac{1}{n}\sum_{i=1}^{n} E_{\ell,a_1,a_2,j}(x_i)\right| \leq \sqrt{\frac{\ln(6kA|S||\mathcal{L}|/\delta)}{n}}$$

for all $a_1, a_2 \in \mathcal{A}$ with $a_1 \neq a_2$, all $\ell \in \mathcal{L}$, and predictor $j \in [k]$. Combining this with the bound on $\frac{1}{n}\sum_{i=1}^{n} E_{\ell,a_1,a_2,j}(x_i)$ given by the stopping condition, we can obtain the fourth guarantee. $\qquad\square$

Note that, with the fourth condition in Theorem E.1, we can also obtain that the size of the disagreement event between other predictors, $f_i, f_j$ with $i, j \neq 1$ satisfies

$$\mu(E^{2\alpha}_{a_1,a_2,\ell}(f_i(x), f_j(x), x)) \leq 2\eta A.$$

For each $x \in E^{2\alpha}_{a_1,a_2,\ell}(f_i(x), f_j(x), x))$, it must be either $x \in E^{\alpha}_{a_1,a,\ell}(f(x), f_i(x), x)$ or $x \in E^{\alpha}_{a_1,a,\ell}(f(x), f_j(x), x)$ for some $a \in \mathcal{A}$. Therefore,

$$\mu(E^{2\alpha}_{a_1,a_2,\ell}(f_i(x), f_j(x), x))$$
$$\leq \sum_{a \in \mathcal{A} \setminus \{a_1,a_2\}} \left( \mu(E^{\alpha}_{a_1,a_2,\ell}(f(x), f_i(x), x)) + \mu(E^{\alpha}_{a_1,a_2,\ell}(f(x), f_j(x), x)) \right)$$
$$\leq \sum_{a \in \mathcal{A} \setminus \{a_1,a_2\}} 2\eta \leq 2A\eta.$$

# F    EXTENSION TO INFINITELY LARGE ACTION SET

We extend our results to a setting where the action space $\mathcal{A}$ of the downstream decision-making tasks is infinitely large. Now that we cannot enumerate all actions in the action space, the disagreement event in this setting can not specify the best-response action. For a loss $\ell \in \mathcal{L}$, one approach is to define the event as

$$E^{\alpha}_{\ell}(f_1(x), f_2(x), x) = \mathbb{I}\big[ \langle f_1(x), \, \ell_{\pi^{\mathrm{BR}}(f_2(x))} - \ell_{\pi^{\mathrm{BR}}(f_1(x))} \rangle > \alpha, or$$
$$\langle f_2(x), \, \ell_{\pi^{\mathrm{BR}}(f_1(x))} - \ell_{\pi^{\mathrm{BR}}(f_2(x))} \rangle > \alpha \big].$$

We now want to "patch" a predictor on the disagreement event, such that the predictor's Brier score decreases by a large amount after patching. However, using this definition of disagreement event, we cannot guarantee that at least one of $f_1$ and $f_2$ has a large prediction error. For example, consider $y_1 = y_2 = [1/2, 1/2]$, predictors $f_1(x_1) = f_2(x_2) = [1, 0]$, $f_1(x_2) = f_2(x_1) = [0, 1]$, and the loss $\ell_a \in \{[x, 1-x] : x \in [0, 1]\}$. According to $f_1$, the loss vector of the best-response action of $x_1$ is $\ell_{a_1} = [0, 1]$, and that of $x_2$ is $\ell_{a_2} = [1, 0]$. According to $f_2$, the loss vector of the best-response action of $x_1$ and $x_2$ is $\ell_{a_2}$ and $\ell_{a_1}$, respectively. Clearly, data $x_1$ and $x_2$ are both in the disagreement event $E^{\alpha}_{\ell}$ for any $\alpha < 1$, but the mean estimation error of $f_1$ and $f_2$ on the disagreement event $\{x_1, x_2\}$ are both 0, since $1/2([0,1] + [1,0]) = [1/2, 1/2]$.

A solution to this problem is to further separate the $E^{\alpha}_{\ell}$ into $2d$ sub-events that focus on the disagreement per-label. The union of these $2d$ sub-events is equivalent to $E^{\alpha}_{\ell}$, as shown in Lemma F.3. Formally, we have the following definition.

**Definition F.1** (Per-label Disagreement Event). *For each a loss $\ell \in \mathcal{L}$, index $i \in [d]$, we define*

$$E^{\alpha}_{\ell,i,>}(f_1(x), f_2(x), x) = \mathbb{I}\big[ f_1(x)_i - f_2(x)_i > \alpha/d, \quad E^{\alpha}_{\ell}(f_1(x), f_2(x), x) = 1 \big].$$
$$E^{\alpha}_{\ell,i,<}(f_1(x), f_2(x), x) = \mathbb{I}\big[ f_1(x)_i - f_2(x)_i < -\alpha/d, \quad E^{\alpha}_{\ell}(f_1(x), f_2(x), x) = 1 \big].$$

To ensure the predictor's expected loss is accurate, we define a per-label best-response event using a discretized approach, approximating each index of the loss vector $\ell_a$ with a finite grid $[1/\mathcal{T}]$ (Definition D.1). Then, if a predictor is approximately accurate with respect to each per-label best-response event, then its estimated loss for the best-response actions are approximately accurate.

**Definition F.2** (Per-label Best-Response Event). *Given a predictor $f$, a loss $\ell$, and a level $\tau \in [1/\mathcal{T}]$, for each index $i \in [d]$, we define the best-response event as*

$$E_{i,\tau}(f(x), x) = \mathbb{I}\{x : \tau - 1/\mathcal{T} < \ell_{\pi^{\mathrm{BR}}(f(x)),i} \leq \tau\}.$$

*where $\ell_{\pi^{\mathrm{BR}}(f(x)),i}$ denote the $i$-th index of $\ell_{\pi^{\mathrm{BR}}(f(x))}$.*

With the disagreement and best-response events now clearly defined, we can adapt our existing algorithm (Algorithm 2) to this new context: at each iteration, we find the largest per-label disagreement events and patch on the predictor with the larger estimation error; then, in a subroutine similar to Algorithm 1, we iteratively calibrate the predictor we just patched on, using the per-label best-response events. This algorithm can maintain similar guarantees as Theorem 3.5, with a data size $\tilde{\Omega}\left( d^2 / (\eta^2 \beta \min\{\beta^2, \eta\alpha^2/d^3\}) \right)$. The formal algorithm is written in Algorithm 6.

Given our approximation of each action feature $\ell_a$ using a finite grid $[1/\mathcal{T}]$, it may seem intuitive to approximate the entire infinite action set as $[1/\mathcal{T}]^d$ and simply run Algorithm 2. However, such an approach yields an action set whose size grows exponentially with $d$, which in turn would require a data sample size that also grows exponentially to achieve guarantees comparable to those in Theorem 3.5. In contrast, our algorithm 6 operates with a polynomial data requirement relative to $d$, making it more feasible for practical application.

---

**Algorithm 5:** Decision Calibration for Infinite Action Set

---

**Input:** Predictor $f$, loss family $\mathcal{L}$, $\beta > 0$, event $E$
1: Let $f^0 = f$.
2: **while** $f^t$ is not index-wise $\beta$-multicalibrated with respect to events $E_{\ell,i,\tau} \cap E$ for some $\ell \in \mathcal{L}$, $i \in [d]$, and $\tau \in [1/\mathcal{T}]$ **do**
3:      Let $\ell^t, i^t, \tau^t = \arg\max_{\ell,i,\tau} \|\mathbb{E}_{(x,y)\sim\mathcal{D}}[(y - f^t(x))E_{\ell,i,\tau}(f^t(x), x)]\|_2$
4:      Let $\phi^t = \mathbb{E}_{(x,y)\sim\mathcal{D}}[y - f^t(x)|E_{\ell^t,i^t,\tau^t}(f^t(x), x) = 1]$
5:      Patch $f^{t+1}(x) = \text{proj}_{[0,1]^d}(f^t(x) + \phi^t E_{\ell^t,i^t,\tau^t}(f^t(x), x))$
6:      $t = t + 1$.
**Output:** $f^t$

---

---

**Algorithm 6:** Reconcile Decision Calibration for Infinite Action Set (ReDCal-Inf)

---

**Input:** $f_1, f_2, \mathcal{L}, \eta > 0, \alpha > 0, \beta > 0$
1: Let $f_1^0 = f_1, f_2^0 = f_2$ and $t = 0$.
2: **while** $\mu(E_\ell) \geq \eta$ for some $\ell \in \mathcal{L}$ **do**
3:      Let $\ell^t, j^t, \cdot^t = \arg\max_{\ell,j,\cdot} \mu(E_{\ell,j,\cdot})$, $E^t = E_{\ell^t,j^t,\cdot^t}$.
4:      Pick

$$i^t = \arg\max_{i \in \{1,2\}} \big| \mathbb{E}_{(x,y)\sim\mathcal{D}}[\ell^t(y, \pi^{\mathsf{BR}}(f_1^t(x))) - \ell^t(y, \pi^{\mathsf{BR}}(f_2^t(x)))|E^t(x) = 1]$$

$$- \mathbb{E}_{x\sim\mathcal{X}}[\ell^t(f_i(x), \pi^{\mathsf{BR}}(f_1^t(x))) - \ell^t(f_i(x), \pi^{\mathsf{BR}}(f_2^t(x)))|E^t(x) = 1]\big|.$$

5:      Denote $f_{i^t}^t$ as $f_i^t$. Let $\phi^t = \mathbb{E}_{(x,y)\sim\mathcal{D}}[y|E^t(x) = 1] - \mathbb{E}_{x\sim\mathcal{D}_\mathcal{X}}[f_i^t(x)|E^t(x) = 1]$.
6:      Patch $f^t(x) = \text{proj}_{[0,1]^d}(f_i^t(x) + \phi^t E^t(x))$.
7:      Let $f_i^{t+1} = \text{Decision-Calibration-Inf}(f^t, \mathcal{L}, \beta, E^t)$. $t = t + 1$.
**Output:** $f_1^t, f_2^t$

---

We now present the formal proofs related to Algorithm 6.

**Lemma F.3.** *For any loss $\ell \in \mathcal{L}$, we have*

$$E_\ell^\alpha = \bigcup_{i\in[d],\cdot\in\{>,<\}} E_{\ell,i,\cdot}^\alpha.$$

*Proof.* By definition, we may get

$$\bigcup_{i\in[d],\cdot\in\{>,<\}} E_{\ell,i,\cdot}^\alpha \subseteq E_\ell^\alpha.$$

For the other direction, assume for the sake of contradiction that there is an element $x \in E_\ell^\alpha$ such that $x \notin E_{\ell,i,\cdot}^\alpha$ for all $i \in [d]$ and $\cdot \in \{>,<\}$. Then, this means

$$|f_1(x)_i - f_2(x)_i| < \alpha/d$$

for all $i \in [d]$. We can then get the inequality

$$\langle f_1(x) - f_2(x), \ell_{\pi^{\mathsf{BR}}(f_2(x))} - \ell_{\pi^{\mathsf{BR}}(f_1(x))} \rangle = \sum_{i\in[d]} (f_1(x)_i - f_2(x)_i) \cdot (\ell_{\pi^{\mathsf{BR}}(f_2(x))} - \ell_{\pi^{\mathsf{BR}}(f_1(x))})$$

$$\leq \sum_{i\in[d]} |f_1(x)_i - f_2(x)_i| \cdot |\ell_{\pi^{\mathsf{BR}}(f_2(x))} - \ell_{\pi^{\mathsf{BR}}(f_1(x))}|$$

$$< d \cdot \alpha/d = \alpha.$$

On the other hand, by definition of event $E_\ell^\alpha$, we have for all $x \in E_\ell^\alpha$,

$$\langle f_1(x) - f_2(x), \ell_{\pi^{\mathrm{BR}}(f_2(x))} - \ell_{\pi^{\mathrm{BR}}(f_1(x))} \rangle > \alpha,$$

which leads to a contradiction. $\qquad\square$

The following lemma shows that, if $f$ is $\beta$-index-wise decision-calibrated, then the loss estimation is approximately accurate.

**Lemma F.4.** *If the predictor $f$ is $\beta$-index-wise decision-calibrated, i.e. for all $\ell \in \mathcal{L}$, $i \in [d]$, $\tau \in [1/\mathcal{T}]$,*

$$\mathbb{E}_{(x,y)\in\mathcal{D}}[\|y - f(x)\|_2 \cdot E_{\ell,i,\tau}(f(x), x)] \leq \beta,$$

*then, the loss estimation given by $f$ is approximately accurate for all $\ell \in \mathcal{L}$:*

$$\mathbb{E}_{(x,y)\in\mathcal{D}}[\langle y, \ell_{\pi^{\mathrm{BR}}(f(x))} \rangle] - \mathbb{E}_{(x,y)\in\mathcal{D}}[\langle f(x), \ell_{\pi^{\mathrm{BR}}(f(x))} \rangle] \leq d\mathcal{T}\beta + d/\mathcal{T}.$$

*Letting $\mathcal{T} \geq 1/\sqrt{\beta}$, we get*

$$\mathbb{E}_{(x,y)\in\mathcal{D}}[\langle y, \ell_{\pi^{\mathrm{BR}}(f(x))} \rangle] - \mathbb{E}_{(x,y)\in\mathcal{D}}[\langle f(x), \ell_{\pi^{\mathrm{BR}}(f(x))} \rangle] \leq 2d\sqrt{\beta}.$$

*Proof.* We can rewrite the expected loss as

$$\mathbb{E}_{(x,y)\sim\mathcal{D}}[\langle y, \ell_{\pi^{\mathrm{BR}}(f(x))} \rangle] = \sum_{i\in[d]} \mathbb{E}_{(x,y)\sim\mathcal{D}}[y_i \cdot \ell_{\pi^{\mathrm{BR}}(f(x)),i}]$$

$$\text{(here } \ell_{\pi^{\mathrm{BR}}(f(x)),i} \text{ represents the } i\text{-th index of } \ell_{\pi^{\mathrm{BR}}(f(x))})$$

$$= \sum_{i\in[d]} \sum_{\tau\in[1/\mathcal{T}]} \mathbb{E}_{(x,y)\sim\mathcal{D}}[y_i \cdot \ell_{\pi^{\mathrm{BR}}(f(x)),i} \cdot E_{\ell,i,\tau}(f(x), x)]$$

$$\leq \sum_{i\in[d]} \sum_{\tau\in[1/\mathcal{T}]} \tau \cdot \mathbb{E}_{(x,y)\sim\mathcal{D}}[y_i \cdot E_{\ell,i,\tau}(f(x), x)].$$

Similarly, we can rewrite the estimated loss as

$$\mathbb{E}_{(x,y)\sim\mathcal{D}}[\langle f(x), \ell_{\pi^{\mathrm{BR}}(f(x))} \rangle] > \sum_{i\in[d]} \sum_{\tau\in[1/\mathcal{T}]} (\tau - 1/\mathcal{T}) \cdot \mathbb{E}_{(x,y)\sim\mathcal{D}}[f(x)_i \cdot E_{\ell,i,\tau}(f(x), x)].$$

The estimation error is

$$\mathbb{E}_{(x,y)\in\mathcal{D}}[\langle y, \ell_{\pi^{\mathrm{BR}}(f(x))} \rangle] - \mathbb{E}_{(x,y)\in\mathcal{D}}[\langle f(x), \ell_{\pi^{\mathrm{BR}}(f(x))} \rangle]$$

$$= \sum_{i\in[d]} \sum_{\tau\in[1/\mathcal{T}]} \tau \cdot \mathbb{E}_{(x,y)\sim\mathcal{D}}[y_i \cdot E_{\ell,i,\tau}(f(x), x)]$$

$$- \sum_{i\in[d]} \sum_{\tau\in[1/\mathcal{T}]} (\tau - 1/\mathcal{T}) \cdot \mathbb{E}_{(x,y)\sim\mathcal{D}}[f(x)_i \cdot E_{\ell,i,\tau}(f(x), x)]$$

$$= \sum_{i\in[d]} \sum_{\tau\in[1/\mathcal{T}]} \tau \cdot \mathbb{E}_{(x,y)\sim\mathcal{D}}[(y_i - f(x)_i) \cdot E_{\ell,i,\tau}(f(x), x)]$$

$$+ \sum_{i\in[d]} \sum_{\tau\in[1/\mathcal{T}]} 1/\mathcal{T} \cdot \mathbb{E}_{(x,y)\sim\mathcal{D}}[f(x)_i \cdot E_{\ell,i,\tau}(f(x), x)].$$

Since $f$ is $\beta$-index-wise decision-calibrated, we have

$$\mathbb{E}_{(x,y)\in\mathcal{D}}[|y_i - f(x)_i| \cdot E_{\ell,i,\tau}(f(x), x)] \leq \beta.$$

Therefore,

$$\mathbb{E}_{(x,y)\in\mathcal{D}}[\langle y, \ell_{\pi^{\mathrm{BR}}(f(x))} \rangle] - \mathbb{E}_{(x,y)\in\mathcal{D}}[\langle f(x), \ell_{\pi^{\mathrm{BR}}(f(x))} \rangle] \leq d\mathcal{T}\beta + d/\mathcal{T}$$

$\qquad\square$

We are now ready to give the formal guarantees of Algorithm 6.

**Theorem F.5.** *Fix any distribution $\mathcal{D}$ and dataset $D \sim \mathcal{D}$ containing $n$ samples drawn i.i.d from $\mathcal{D}$. For any pair of predictors $f_1, f_2 : \mathcal{X} \to [0,1]^d$, family of loss functions $\mathcal{L}$, loss margin $\alpha > 0$, disagreement region mass $\eta > 0$, and decision-calibration tolerance $\beta > 0$, Algorithm 6 run over the empirical distribution $D$ updates $f_1$ and $f_2$ for $T_1$ and $T_2$ time-steps, respectively, and outputs a pair of predictors $(f_1^T, f_2^T)$ such that, with probability at least $1 - \delta$ over the randomness of $D \sim \mathcal{D}^n$,*

1. *The total number of time-steps for Algorithm 6 and Algorithm 5 is*

$$T = T_1 + T_2 \leq \frac{2d}{\min\{\beta^2, \eta\alpha^2/2d^3\}}$$

2. *For $i \in \{1, 2\}$, the Brier scores of the final models are lower than that of the input models:*

$$B(f_i^{T_i}, \mathcal{D}) \leq B(f_i, \mathcal{D}) - (T_i - \eta) \cdot \min\left\{\beta^2, \eta\alpha^2/(2d^3)\right\}$$

3. *For $i \in \{1, 2\}$ and for all $\ell \in \mathcal{L}$, the downstream decision-making losses of the final models do not increase by much compared to that of the input models:*

$$\mathbb{E}_{(x,y)\sim\mathcal{D}}[\ell(y, \pi_\ell^{\mathsf{BR}}(f_i^T(x)) - \ell(y, \pi_\ell^{\mathsf{BR}}(f_i(x))] \leq 2T_i\sqrt{\beta}d$$

4. *The final models approximately agree on their best-response actions almost everywhere. That is, the disagreement region $E_\ell^\alpha$ calculated using $f_1^T, f_2^T$ has small mass: $\forall \ell \in \mathcal{L}$,*

$$\mu(E_\ell^\alpha) \leq 2\eta$$

*if $n \geq \Omega\left(d^2/(\eta^2\beta\min\{\beta^2, \eta\alpha^2/d^3\}) \cdot \left(\ln(d\mathcal{T}|\mathcal{L}|) - \ln(\delta\min\{\beta^2, \eta\alpha^2/d^3\})\right)\right)$.*

*Proof.* **Number of time-steps**. By Lemma C.2, at each iteration $t$ in Algorithm 6 of its subroutine Algorithm 1, the Brier score of $f^t$ decreases by $\min\{\beta^2, \eta\alpha^2/2d^3\}/2$, when $m \geq \lceil\sqrt{d/2\min\{\beta^2, \eta\alpha^2/2d^3\}}\rceil$. Since Brier score is bounded in $[0, d]$, the total number of time-steps satisfies

$$T_{\max} = T_1 + T_2 \leq \frac{4d}{\min\{\beta^2, \eta\alpha^2/2d^2\}}.$$

**Number of Output Predictors.** We count the number of pairs of output predictors. The patching event at each iteration is in the set $\{E_{\ell,i,\cdot} : \ell \in \mathcal{L}, i \in [d], \cdot \in \{<, >\}\} \cup \{E_{\ell,i,\cdot} \cap E_{\ell',i',\tau} : \ell, \ell' \in \mathcal{L}, i, i' \in [d], \tau \in \mathcal{T}, \cdot \in \{<, >\}\}$. The total number of event is at most $2|\mathcal{L}|d + 2|\mathcal{L}|^2 d^2|\mathcal{T}|$. Since each pair of output predictors is defined by the sequence of patchings and $\Delta^t \in [1/m]^d$, we have the total number of possible output predictors is

$$|S| \leq ((4d^2|\mathcal{L}|^2|\mathcal{T}|)(m+1)^d)^{T_{\max}+1}.$$

Therefore,

$$\ln(|S|) \leq \Omega\left(\frac{d^2}{\min\{\beta^2, \eta\alpha^2/2d^2\}} \ln(d|\mathcal{L}||\mathcal{T}|/\min\{\beta^2, \eta\alpha^2/2d^2\})\right).$$

**Brier Score.** The result directly follows using the same argument as in Theorem 3.5. We can obtain

$$B(f_i^{T_i}, \mathcal{D}) \leq B(f_i, \mathcal{D}) - T_i \cdot \min\left\{\beta^2, \eta\alpha^2/(2d^3)\right\} + \sqrt{(d\ln(6d|S|/\delta)/(2n))}.$$

**Loss Estimation.** Using Lemma D.5 and a union bound over $\{E_{\ell,i,\cdot} \cap E_{\ell',i',\tau} : \ell, \ell' \in \mathcal{L}, i, i' \in [d], \tau \in [1/\mathcal{T}], \cdot \in \{<, >\}\}$ and $|S|$, we have with probability at least $1 - \delta/3$,

$$\left\|\mathbb{E}_{\mathcal{D}}[(y - f(x))E_{\ell,a}(f(x), x)] - \frac{1}{n}\sum_{i=1}^n[(y_i - f(x_i))E_{\ell,a}(f(x_i), x_i)]\right\|_2 \leq \sqrt{\frac{3d\ln(6d\mathcal{T}|S||\mathcal{L}|/\delta)}{2n}}.$$

Using Lemma F.4 and summing over all time-steps, we can obtain the result following the same steps as the proof in Theorem 3.5:

$$\mathbb{E}_{(x,y)\sim\mathcal{D}}[\ell(y, \pi^{\mathsf{BR}}(f_i^T(x)) - \ell(y, \pi^{\mathsf{BR}}(f_i(x))] \leq \left(\sqrt{\beta}d + \sqrt{(3d\ln(6d\mathcal{T}|S||\mathcal{L}|/\delta)/(2n))}\right)T_i$$

**Disagreement Event.** Using union bound over the disagreement events $\{E_\ell : \ell \in \mathcal{L}, \}$ and the set of possible output predictors, we have, for all $\ell \in \mathcal{L}$, and $(f_1, f_2) \in S$,

$$\left| \mu(E_{\ell,a_1,a_2}(f_1(x), f_2(x), x) - \frac{1}{n}\sum_{i=1}^{n} \mathbb{I}[E_{\ell,a_1,a_2}(f_1(x_i), f_2(x_i), x_i))] \right| \leq \sqrt{\frac{\ln(6|S|\,|\mathcal{L}|\,/\delta)}{2n}}.$$

From the while loop condition in Algorithm 6, we know $\frac{1}{n}\sum_{i=1}^{n} \mathbb{I}[E_\ell(f_1^{T_1}(x_i), f_2^{T_2}(x_i), x_i)] \leq \eta$. Therefore,

$$\mu(E_\ell) \leq \eta + \sqrt{(\ln(6|S|\,|\mathcal{L}|\,/\delta)/(2n)} \quad \text{for all } \ell \in \mathcal{L}$$

The theorem follows from solving $n$ in the conditions 2 through 4. $\qquad\square$

## G  ADDITIONAL EXPERIMENT DETAIL

Our ImagineNet experiments are run on a Macbook Pro with 32GB of RAM. The experiment on the ImageNet dataset uses pre-trained models provided by pyTorch, which do not require additional training. The experiment on the HAM10000 dataset includes neural network models trained using pyTorch on NVIDIA GA100 GPU (80 GB of RAM) with 2 compute workers loading the data. The total time to train the two neural network models with approximately $88\%$ Top-1 accuracy takes less than 5 minutes.

### G.1  ADDITIONAL EXPERIMENT ON IMAGENET DATASET

**Brier score.** In Figure 4, we compare the Brier score of ReDCal (Algorithm 2) with the two baseline algorithms on both the calibration and the test datasets. Compared to Reconcile, our algorithm decreases the Brier score by a smaller amount on the test dataset. The combined algorithm of Decision-Calibration with ReDCal as post-process achieves the most substantial decrease in the Brier score.

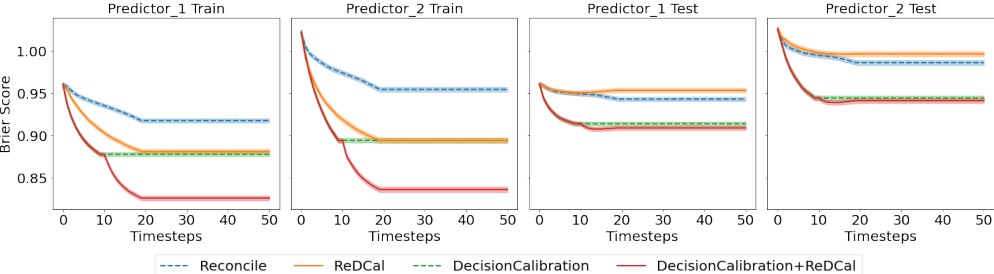

Figure 4: ReDCal decreases Brier score on Imagenet. Compared to Reconcile, our algorithm decreases the Brier score by a smaller amount on the test dataset. Decision-Calibration with ReDCal as post-process achieves the most substantial decrease in the Brier score.

**Decision loss on calibration dataset.** In Figure 5, we compare the decision gap of our proposed algorithm with the two baseline algorithms on the training dataset.

**Decision loss comparison for high-dimensional classification problem.** In Figure 6, we compare the decision loss gap of our proposed algorithm with the two baseline algorithms on the testing dataset, using $d = 10, 100$, and 1000 classes. We plot the average loss gap of the two predictors. The hyperparameters are: disagreement margin $\alpha = 0.1/d$, decision-calibration tolerance $\beta = 0.001/d$, disagreement region mass $\eta = 0.01$, number of actions $K = 10$.

### G.2  HAM1000 CLASSIFICATION

**Experiment Setup.** We use the HAM10000 dataset (Tschandl et al., 2018) (licensed CC BY-NC 4.0) on pigmented skin lesions to predict the probability that a patient has contracted one of

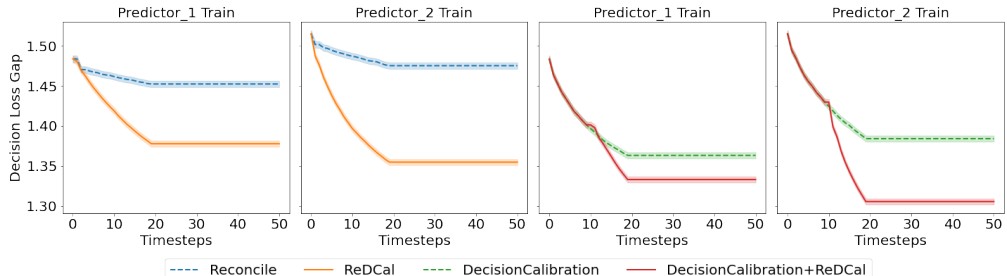

Figure 5: ReDCal decreases decision loss on Imagenet. We plot the gap between the average loss of each predictor and the optimal loss had we known the true predictor $f^*$ on the calibration set. **Left two figures:** We compare the LossGap of ReDCal (orange) with Reconcile (blue). Compared to Reconcile, our algorithm converges at a similar rate and decreases the loss by a larger amount on the test dataset. **Right two figures:** We compare the LossGap of Decision-Calibration (green) to that of Decision-Calibration with a run of ReDCal as post-process (red). Our proposed algorithm can improve upon Decision-Calibration output predictors and further reduce the loss on the test dataset. Results are averaged over 500 runs and the shaded region indicates $\pm 1$ standard errors.

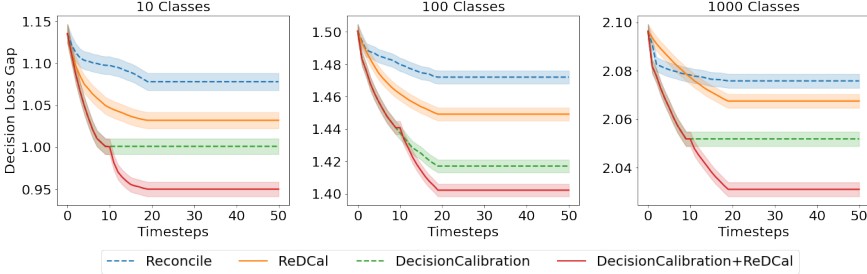

Figure 6: ReDCal decreases decision loss on Imagenet. The takeaway results are similar to Figure 2. As the number of classes in the multi-class classification problem grows from 10 to 1000, ReDCal still outperforms Reconcile in decreasing decision loss on the test dataset. When we have 1000 classes, ReDCal converges slower than Reconcile. Furthermore, ReDCal can further decrease the decision loss when it is used as a post-process after Decision Calibration terminates.

7 possible skin diseases: 'akiec', 'bcc', 'bkl', 'df', 'nv', 'vasc', and 'mel'. We split the dataset into train/validation/test sets, with 20% of the data are used for validation and 20% are used for testing. We use the train set to train two neural networks using pyTorch with resnet50 (He et al., 2015) and densenet121 (Huang et al., 2018) architectures and learn two models with around 88% top-1 accuracy. From each model, we output the individual probability prediction for each of the 7 possible labels. We use the validation set to calibrate the predictors using our proposed algorithm and the two baseline algorithms, and the test set to measure the final performance.

We run each calibration algorithm 10 times. At each run, we draw a fresh loss function created based on the loss function motivated by medical domain knowledge in Zhao et al. (2021) and additional random noise drawn from $\mathsf{Normal}(0, 1)$. There are two possible actions for the decision-maker: treatment ($a = [1, 0]$) or no treatment ($a = [0, 1]$). Given a loss function $\ell$ and a predictor $f$, the decision-maker will choose an action that minimizes their loss.

For each calibration algorithm, we calculate (1) the Brier score of the updated predictors and (2) the differences between the optimal loss had we known $y$ and the actual loss from taking the best-response actions induced by each predictor.

The hyperparameters for Algorithm 2 are chosen as follows: loss margin $\alpha = 0.1$, target disagreement region mass $\eta = 0.01$, and decision-calibration tolerance $\beta = 0.000001$.

**Results.** The takeaway results are similar to those of the ImageNet experiment.

**Brier score.** In Figure 7, we compare the Brier score of ReDCal (Algorithm 2) with the two baseline algorithms on both the calibration and the test datasets. Compared to Reconcile, our proposed algorithm decreases the Brier score by a smaller amount.

**Decision loss.** In Figure 8, we compare the decision loss gap of our proposed algorithm with the two baseline calibration algorithms. Compared to Reconcile, our algorithm decreases the decision loss by a larger amount on the test dataset. Furthermore, while Decision-Calibration already decreases the decision loss, our algorithm can further improve upon their result when it is used as a post-process after Decision-Calibration terminates.

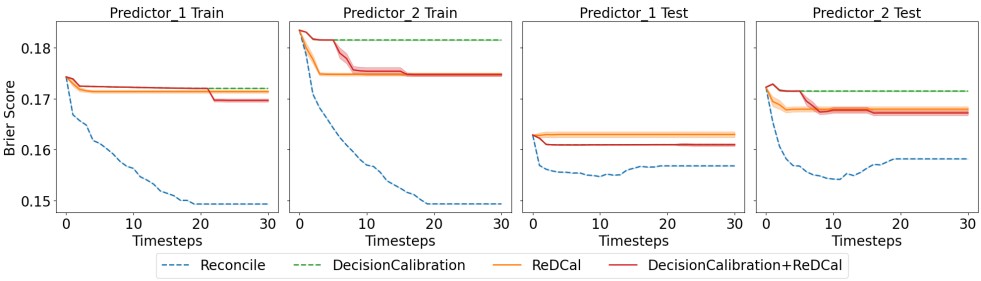

Figure 7: Brier score of the updated predictors using Algorithm 2 (orange) and two benchmark algorithms: Algorithm 3 (dashed-blue) and Algorithm 1 (dashed-green). Our algorithm reduces the Brier score by a smaller amount compared to Algorithm 2. Results are averaged over 10 runs and the shaded region indicates ±1 standard error.

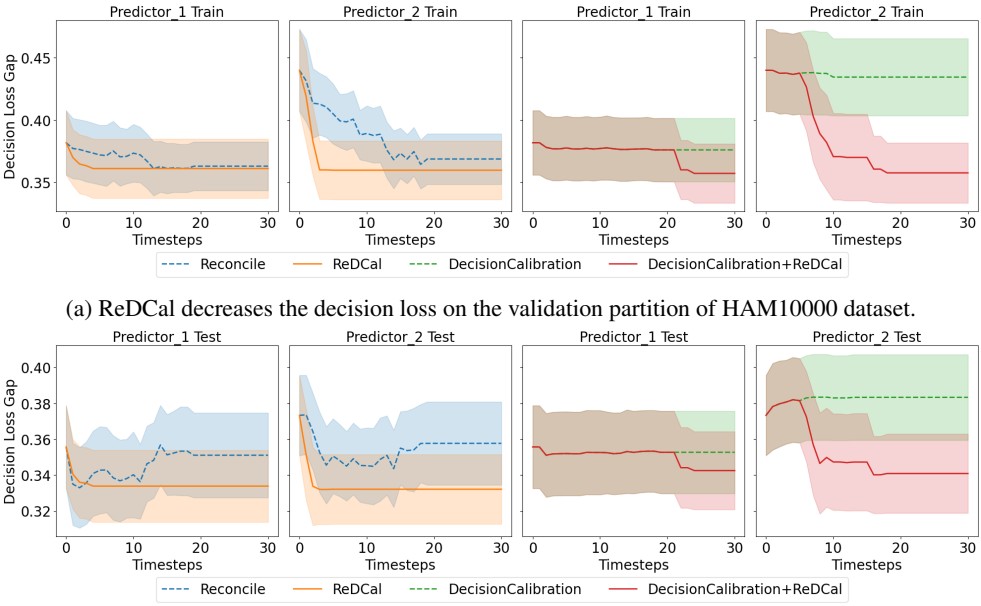

(a) ReDCal decreases the decision loss on the validation partition of HAM10000 dataset.

(b) ReDCal decreases the decision loss on the test partition of HAM10000 dataset.

Figure 8: In Figure 8a and Figure 8b, we plot the gap between optimal loss had we know the true label $y$ and the loss from taking best-response actions induced by the calibrated predictors on the validation set and test set, respectively. In the left two figures, we compare Algorithm 1 (orange) with Algorithm 3 (blue). While the average loss of predictors updated using Algorithm 3 may increase on the test set, our algorithm quickly converges and produces predictors with lower decison-making loss. In the right two figures, we compare Algorithm 1 (green) to Algorithm 1 with an additional run of Algorithm 2 (red) as post-process. We observe that running our algorithm as post-process can still further decrease the loss compared to just running Algorithm 1 on its own. Results are averaged over 10 runs and the shaded region indicates ±1 standard errors.

