# OpenReview forum: "Reconciling Model Multiplicity for Downstream Decision Making"
_ICLR.cc/2025/Conference — ICLR 2025 Poster_

### Official Review · Reviewer_x3z4 · 2024-11-03

**Soundness:** 3
**Presentation:** 3
**Contribution:** 3
**Rating:** 6
**Confidence:** 3

**Summary:**

The paper provides a framework using multi-calibration to reconcile model multiplicity in downstream decisions. It seeks to develop a framework to address the inherent discrepancies between predictive models, which often result in varying decision recommendations despite equivalent accuracy levels. The authors propose an algorithm that aligns predictive models with decision-making, improving agreement on best-response actions and losses. Empirical validation shows enhancements in downstream decision performance and addressing prediction disagreements, relevant to multi-calibration advancements. Proposed approach improves the consistency and utility of models in decision-based applications, including cases where only empirical data is available.

**Strengths:**

The paper presents a novel framework to address the issue of model multiplicity in predictive modeling for decision-making, using multi-calibration techniques. It identifies specific hyperparameters, such as loss margin and decision-calibration tolerance as the driving parameters of key results. In practice, these parameters will need to be adjusted to mediate trade-offs between model fairness and computational efficiency. The authors provide both theoretical insights and empirical validations of their approach. The paper makes a reasonable contribution by integrating theoretical multi-calibration concepts into empirical settings and testing them across multiple datasets.

**Weaknesses:**

Comments/Questions to Authors:
•	The initial concept of multi-calibration was introduced by Hebert-Johnson et al. in 2017 which focused on ensuring fairness across overlapping subpopulations. Recent advancement includes extension of multi-calibration to game dynamics and multi-objective learning – for example, work by Nika Haghtalab and Eric Zhao, which utilize game dynamics to connect fairness and optimization. Another line of work by Roth et al deals with a wide array of issues from online multicalibrated learning to omnipredictors. While the article does a good job at comparing and showing differences with respect to work of Roth, the conceptual difference with respect to work of Haghtalab et al. would be much appreciated.

•	The algorithm’s performance depends on several hyperparameters (such as loss margin and decision-calibration tolerance). It seems techniques to fine-tune parameters such as loss margin and decision-calibration tolerance are crucial for achieving the results and trade-offs that is important in practice (e.g., between prediction fairness and accuracy). These are critical for adapting multi-calibration models to robustly function across heterogeneous data distributions and complex decision environments, the paper should clearly discuss how the choice of these hyperparameters impact the performance of proposed algorithm. I am also curious about the third result of Theorem 3.2 (remark 3.3). How does the impact of this result show up in practice/experiments? I believe a major limitation of the paper is lack of explicit focus on loss margin and decision-calibration tolerance hyperparameters.

•	The paper could better motivate the empirical tests conducted and discuss how they confirm the theoretical claims and show improvements in decision-making outcomes and model agreement. Avenues for future work on validating the proposed algorithm's effectiveness in real-world data scenarios should be discussed.

•	Additional literature – please clarify the relationships / novelty of this paper with respect to work of these authors. I would be especially interested in knowing the relevance of this paper with issues of fairness-accuracy tradeoff.

“Inference for an Algorithmic Fairness-Accuracy Frontier,” Yiqi Liu and Francesca Molinari (2024). This work provides a consistent estimator for a fairness-accuracy frontier. Method for testing fairness-related hypotheses in algorithmic decision-making seems relevant.

“Fair Representation: Guaranteeing Approximate Multiple Group Fairness for Unknown Tasks,” Xudong Shen and Yongkang Wong and Mohan S. Kankanhalli, IEEE Transactions on Pattern Analysis and Machine Intelligence (2021). Explores approximate fairness using fair representation across multiple fairness notions.

“Multigroup Robustness,” Lunjia Hu and Charlotte Peale and Judy Hanwen Shen (2024). This work establishes a connection between multigroup fairness and robustness and discusses robustness algorithms tailored for subpopulations under data corruption.

“Inherent Trade-Offs in the Fair Determination of Risk Scores,” J. Kleinberg and S. Mullainathan and Manish Raghavan (2016). Shows inherent trade-offs in algorithmic fairness in risk scores and formalizes three fairness conditions and proves their incompatibility in most cases.

“Multi-group Agnostic PAC Learnability,” G. Rothblum and G. Yona, International Conference on Machine Learning (2021). Provides a framework for multi-group agnostic PAC learning.

Also important is work by Michael P. Kim et al. on Multiaccuracy: Black-Box Post-Processing for Fairness in Classification (2018): It introduces a foundational framework for ensuring fairness in classification, with a focus on multi-group fairness using multiaccuracy post-processing techniques. This paper sets a critical baseline for future research on fairness in predictions.

**Questions:**

Please see my questions above. Additionally:

1) How does the framework perform across different domains or datasets that exhibit varying levels of complexity and noise?
2) As mentioned, determining optimal ranges and conditions for key hyperparameters parameters across varying contexts can be challenging.
3) Although the paper addresses computational efficiency, the scalability of the algorithm for very large datasets remains an open question. How will the computational complexity grow with increased data size?

---

> ### Author Response · Authors · 2024-11-22
>
> We would like to thank the reviewer for their time and insightful comments. Please find our responses below.
>
> **Regarding related work**: We have added the following paragraph to our additional related work section in Appendix A:
>
> Within the literature on multigroup fairness, Kleinberg et al. (2016) highlighted the trade-offs in fairness by proving the incompatibility of three key fairness conditions—calibration, positive class balance, and negative class balance—emphasizing the need for prioritization in fairness criteria. Building on these foundational insights, Kim et al. (2019) introduced multiaccuracy. This post-processing technique ensures consistent accuracy across subpopulations without access to group
> labels, laying the baseline for future work on predictive fairness. Following such framework, Rothblum & Yona (2021) extended the PAC learning paradigm to multi-group scenarios. Shen et al. (2023) proposed a framework for fair representations across multiple fairness notions and unknown tasks. More recently, Hu et al. (2024) explored robustness in multigroup fairness and proposed algorithms tailored for subpopulations under data corruption. Liu & Molinari (2024) proposed a method to estimate the fairness-accuracy frontier for testing fairness in decision-making. Another line of work (Haghtalab et al., 2023) connects fairness and optimization with a unified game theoretic framework. However, our work does not focus on the fairness aspect of multicalibration but instead borrows its techniques.
>
> **Regarding the hyperparameters in Algorithms 1 and 2**: The loss margin $\alpha$ and decision-calibration tolerance $\beta$ are the targets that we are optimizing for in Algorithm 2 and Algorithm 1, respectively. For a larger value of $\beta$ (Definition 2.5), a predictor $f$ is further away from the true label on the set of best-response event $\mathcal{E}$ compared to a smaller value of $\beta$. That is, a smaller value of $\beta$ requires Algorithm 1 to run for more iterations before it can converge, as seen in the first result of Theorem 3.4.
>
> Similarly, the loss margin $\alpha$ is how much we can tolerate the difference in decision-making losses when the two predictors induce different best-response actions in the downstream task. A higher value of $\alpha$ leads to a shorter convergence time for Algorithm 2 (first result) and a worse guarantee of the improvement of the Brier score (second result) in Theorems 3.2 and 3.4.
>
> **Regarding the third result in Theorem 3.2**: The theorem establishes that the decision loss will increase only minimally, if at all. In the experimental results shown in Figures 2 and 3, the decision loss strictly decreases when running ReDCal, which is ideal and confirms the theory.
>
> **Regarding our empirical findings**: In Figures 4 and 7 (Appendix F), we show that our proposed algorithms can reduce the Brier score of the input models (second result in Theorems 3.3 and 3.5). In Figures 2 and 3, we show that our proposed algorithm can decrease the decision loss (third result) by more than the baseline Reconcile algorithm and the Decision Calibration algorithm (when our algorithm is used as post-processing). Also, the final predictor1 and predictor2 have similar loss values, which corresponds to the fourth result.
> Both the Imagenet dataset and the HAM10000 dataset are real-world datasets.
>
> **Regarding the computational complexity**: with a larger dataset and large disagreement region for the base predictors, our algorithms still converge in the same number of time steps as it does not depend on the number of samples. The additional error in Theorem 3.4 compared to Theorem 3.2 goes to zero as we have more data. At each time step, Algorithms 1 and 2 compute a single patching operation (line 5 of Algorithm 1, line 6 of Algorithm 2) to update the current predictor. With a larger disagreement event, this patching operation will update on more individual samples, which should scale with the size of the disagreement event.

---

> > ### Comment · Reviewer_x3z4 · 2024-11-25
> >
> > Thank you for your response - much appreciated. I keep my original score.

---

### Official Review · Reviewer_1e2c · 2024-11-04

**Soundness:** 3
**Presentation:** 3
**Contribution:** 3
**Rating:** 6
**Confidence:** 4

**Summary:**

The paper presents a new algorithm for reconciling two or more models in a supervised setting; the goal is to address the model multiplicity problem (which refers to the phenomenon that there may be > 1 model with similar accuracy but substantial disagreements among predictions) in a decision-aware way. Specifically, the paper shows how to perform model reconciliation in a way that (1) makes the models agree in the prediction space, (2) does not hurt (and possibly even improves) the squared loss of either model, and (3) does not hurt (and possibly even improves) the decision loss of a given downstream decision-maker. (The decision loss is an arbitrary linear loss that the decision maker uses to map predictions to actions.) In addition, the paper performs preliminary experiments showing that the method outperforms (with respect to decision loss) a prior reconciliation algorithm on some vision datasets.

**Strengths:**

This paper is overall a good and novel contribution to the predictive multiplicity literature. Specifically:

+ The contribution of this paper appears new in the model multiplicity literature: it gives a rigorous method for reconciliation of any two models with provable guarantees in terms of the resulting model loss (for which only one algorithm exists in the literature), while at the same time ensuring in a rigorous way that downstream decision making is not affected negatively (which is new); and it moreover extends this method to more than 2 models in a natural way.

+ In fact, from what I can tell, it has an even broader scope than typically considered in the model multiplicity context, as it does not require the to-be-reconciled models to start off having similar or equal accuracy. When the to-be-reconciled models do have similar accuracy to begin with, then the paper's contribution intuitively appears to offer the interesting and thought-provoking semantics of: "given two different models with similar performance, you can bypass the multiplicity issue by finding an even higher-performing model (which ensembles/combines the two initial models), with similar or better downstream decision making guarantees".

**Weaknesses:**

While overall I believe this to be a good-quality paper, there is the following (relatively non-major) consideration that I would call a weakness:

- The paper currently appears written with a primarily theoretical audience in mind, but I think it could still do a better/more thorough job coming up with/describing experiments. It currently gives two semi-synthetic ones. In the first one, linear decision losses are generated in a Gaussian manner --- so that the two vision models are essentially being calibrated to realizations of random noise (granted, the experiment does illustrate the point that the new method achieves better decision-making properties than the old one, but the setup still sounds strange). In the second one, the decision loss is a "loss function motivated by medical domain knowledge in Zhao et al (2021) and additional random noise" --- and in this case, the paper should at the very least clearly state what the loss function in Zhao et al (2021) is, and perhaps provide more principled perturbations of it than Gaussian noise ones.

Neither of these experiments, however, connects in any way to the examples given earlier in the paper on how disregarding decision loss can lead to its deterioration during the reconciliation procedure. These examples are in some sense prototypical of how reconciliation could hurt downstream decision making; and given how easy it is to come up with a synthetic task/models + synthetic loss clearly showcasing them (i.e. making it to where some significant mass of items will flip across the decision boundary as a result of the reconciliation), I would like to urge the authors to do just that as it will make the paper more logically coherent.

**Questions:**

The paper currently appears mostly in good shape; my main small improvement suggestions are about the experimental section (at least by lightly tweaking existing experiments/adding another simple but conceptual one).

In addition, even though the writing of the paper is mostly quite good, it could still be improved in a few places. For instance, the subsection about handling more than 2 models is somewhat confusingly written: e.g. consider the sentence about exponentially many output predictors in the number of models k, followed by the sentence that union and Chernoff-Hoeffding bounds "suggest" that the sample complexity scales linearly in k --- a reader who may not have checked the details in the appendix may easily confuse this for saying that the method may not necessarily be scalable in k, even though it actually is. Also, the event collection setup in that case, which involves reconciling a single base model with every other model, is not described that well in prose --- so consider e.g. including a quick diagram which could e.g. have the base model with "reconciliation arrows" pointing to the other models, or something similar.

---

> ### Author Response · Authors · 2024-11-22
>
> We would like to thank the reviewer for their time and insightful comments. Please find our responses below.
>
> **Regarding the experiment**: The loss function in Zhao et al (2021) assigns a loss value to each of the 7 possible skin diseases and actions (treatment or no treatment). The loss function is described in their Figure 1. Blue indicates low loss, and red indicates high loss. Since the authors did not provide an exact value for this loss function, we map the color to the closest value (as shown in the table below) and add Gaussian noise to emulate their setup.
>
> In Figure 3, the Reconcile algorithm only seeks to reconcile individual probability predictions and does not take into account the downstream decision-making loss. For both predictors f_1 and f_2, we observe that the decision loss gap increases as we run this algorithm for longer. On the other hand, our algorithm reduces the loss gap for both predictors. This indicates that we should not disregard decision loss, aligning with our argument in Theorem 2.7.
>
> |             | akiec |  bcc |  bkl |   df |  nv | vasc |  mel |
> |------------:|------:|-----:|-----:|-----:|----:|-----:|-----:|
> | NoTreatment |  12.5 | 10.5 | -2.0 | -3.0 | 4.0 | -1.5 |  8.5 |
> |   Treatment |  -0.5 | -2.0 |  4.0 | -0.5 | 1.0 |  3.0 | -3.0 |
>
>
> **Reconciling more than 2 models**: We thank the reviewer for their comment on the write-up in Section 3.3. We have rephrased the text.

---

> > ### Comment · Reviewer_1e2c · 2024-11-25
> >
> > Thank you for the update and for the clarification regarding the experiments. With several updates for the sake of readability that have been promised (as per your response to my review and to the other reviews), I believe the paper should be in good shape. I have no unresolved questions regarding this submission and keep my original score.

---

### Official Review · Reviewer_ctoE · 2024-11-04

**Soundness:** 2
**Presentation:** 1
**Contribution:** 2
**Rating:** 6
**Confidence:** 4

**Summary:**

This paper studies the Reconcile algorithm proposed in [1] which takes two prediction models and outputs two models that have better prediction accuracy which is quantified by the Brier score. They expand on the existing method by proposing a new reconcile algorithm that looks to improve prediction accuracy while also preserving the downstream decision-loss. They provide theoretical bounds that show their method trades off improvement to prediction accuracy at the cost of downstream decision-loss.  They provide numerical results to show their algorithm helps reduce the loss gap after running their version of the Reconcile algorithm.



-----
[1] Aaron Roth, Alexander Tolbert, and Scott Weinstein. Reconciling individual probability fore- casts. In Proceedings of the 2023 ACM Conference on Fairness, Accountability, and Trans- parency, FAccT ’23, pp. 101–110, New York, NY, USA, 2023. Association for Computing Ma- chinery. ISBN 9798400701924. doi: 10.1145/3593013.3593980. URL https://doi.org/ 10.1145/3593013.3593980.

**Strengths:**

The paper highlights an important problem, that improvements to prediction models can hurt downstream decision-making since downstream decision-makers may have loss functions that do not necessarily align with prediction accuracy. The paper combines existing work in multi-calibration with work in model multiplicity to solve this problem.  The algorithm proposed by the paper seems novel and provides what seems to be sensible theoretical guarantees that trade-off between improvements to prediction accuracy and preserving decision-loss. This seems to be a relatively significant improvement to the Reconcile algorithm.

**Weaknesses:**

The paper has weaknesses in its presentation as well as results that seem somewhat suspicious/hard to interpret precisely.

The following items could be addressed and improved for presentation:
- In the paper's introduction, the authors mention calibration several times, but for someone not immediately familiar with the literature it's hard to understand what it is formally. It becomes a little better defined at Lemma 2.6, but having extra background or explanation in the introduction would be helpful in understanding the high-level concepts used to construction the paper's proposed algorithm.
- The paper could provide better background on the Reconcile algorithm which seems to be a key motivation for pursuing the work. There is no high level description of how the algorithm works, so Theorem 2.7 for example feels impossible to verify. Additionally, the Reconcile algorithm reproduced in the appendix seems to be missing key elements, like defining $h$ and $\mu$. The former seems to be an important update step.
- The paper does not define notation well. For example $\mu$ and $A$ are not defined or are defined offhandedly, making it hard to decipher when it shows up later in the text.
- In Theorem 2.7 the distribution $\mathcal{D}$ is not defined which makes the last expectation in the proof hard to verify. I could not immediately come up with a distribution where the expectation is 1/2.
- Some notation seems to be incorrect or have a typo. For example the Definition 2.4 of $E_{\ell,a}$ is an indicator outside a set. I don't think this is standard notation and is not precise. My guess is it's a typo.
- Is the notation $\ell_a$ necessary? If so how do you obtain $\ell_a$ given $\ell$?
- The descriptions of Decision-Calibration, ReDCal, and Decision-Calibration + ReDCal are confusing to me. What exactly is the algorithm for the first and last algorithms. When you count the number of time steps for ReDCal, which loop are you counting? Does it include the loops in decision-calibration?

The following items can be addressed to help improve the results of the paper:

**Questions related to Theorem 3.4**
- In Theorem 3.4, does item 1. have a typo?
- Define $A$ the number of actions in the theorem statement for part 3. since it is off-handedly defined somewhere else in the paper.
- How do I choose $\beta$, $\alpha$, and $\eta$ if I wanted to guarantee a certain decision-making loss level while optimizing the Brier scores? Managing the trade-off between prediction accuracy and decision-loss would help improve the impact of this theoretical result. Right now it seems hard since the term $T_i \beta$ seems hard to control.
- The decision gap in the numerical results don't seem to correspond to the bound in 3. Shouldn't it be close to 0 if you choose $\beta = 0.00001$?

**Questions:**

1. One thing I didn't understand was if you are given two models at the end of ReDCal, how do you choose which model to use? Why is it important that the two models agree in the decisions they make if you already control the decision-loss gap?
2. If you don't want to affect the decision-loss, it makes sense to me that you just wouldn't change the two predictors. Is that the case and is it reflected in your algorithm if you choose the correct parameters?
3. Why does the algorithm converge so fast in the numerics? The upper bound on the time steps seems to suggest the number of time steps that should be run before convergence should be large given the choices of $\beta$ and $\alpha$. Does this imply the bound is is potentially very loose and is it possible for it to be made tighter?

---

> ### Author Response · Authors · 2024-11-22
>
> We would like to thank the reviewer for their time and insightful comments. Please find our responses below.
>
> **Regarding Theorem 3.4**: We apologize for the typo in item 1, we have fixed it in our revision. For the questions:
> 1. The trade-off between prediction accuracy and decision loss: In the setting where the distribution $\mathcal{D}$ is given to us (Remark 3.3), we can set $\beta = \alpha / (T \sqrt{d} A)$ to ensure the decision-loss does not increase by more than $\alpha$. There are no trade-offs in this case.
> In a more practical setting where a dataset $D$ is given, as shown in the experiment results in Fig. 2 and Fig. 3, the decision loss consistently decreases with each iteration of ReDCal. This implies that for larger $T_i$​, both prediction accuracy and decision loss generally improve, reducing the need to balance these two factors.
> 2. Decision gap and numerical results: The third item in Theorem 3.4 indicates that the decision loss of the final predictors $f_i^T$ is no larger than the decision loss of the initial predictors $f_i^0$. This does not imply that the decision loss of $f_i^T$ is necessarily close to that of $f_i^0$; rather, a decrease in decision loss is the desired outcome. The numerical results confirm this behavior, showing a reduction in decision loss that aligns with our theoretical findings.
>
> **Regarding the two models at the end of ReDCal**: The decision-maker can choose either of the final models after running ReDCal. Even when the decision-loss gap is controlled, it is important to ensure that the two output models agree on the best-response actions almost everywhere. In our motivating example, a doctor may feel more confident providing treatment to a patient when both of the final models suggest that treatment is the best-response action, compared to a scenario where each model induces a different best-response action.
>
> **Regarding the decision loss**: If the goal is to avoid any changes to the decision loss, the "agreement" between the two predictors becomes irrelevant. In this case, we can set the disagreement region mass $\eta$ to a large value, such as 1. This choice of $\eta$ causes ReDCal to terminate immediately without updating the predictors.
>
> However, in our model multiplicity setting, the objective is to minimize the size of the disagreement region. A smaller disagreement region is desirable because, during a disagreement event, at least one predictor either fails to provide the correct best-response action or yields an inaccurate expected loss. By reducing the disagreement region, we enhance the reliability of both predictors and the decisions they inform.
>
> Does this clarification address your concern about changes in the decision loss?
>
> **Regarding the convergence rate in experiments**: The upper bounds in Theorem 3.2 and 3.4 are worst-case upper bounds. Hence, it is reasonable to have the algorithm converge at a faster rate in numerical experiments.
>
> **Regarding the presentation of the paper**: We thank the reviewer for their suggestion. Below we clarify each of the points and highlight the changes we made to our file.
> 1. Background on calibration: We have added a sentence explaining calibration in the introduction.
> 2. $A$ and $\mu$: For $A$, we have defined in line 210 that it is the number of actions and added it in the theorem statement. We also added the definition of $\mu$ right before section 3.1.
> 3. Distribution in Theorem 2.7: We have already provided the distribution in the first sentence of the proof (around line 284) by saying that $Pr[x=1]=Pr[x=2]=0.5$.
> 4. $\ell_a$: $\ell_a$ is introduced in the first paragraph of Section 2.2 (around line 212). We assume that the decision loss is linear in the label $y$. This assumption implies that there exists a vector $\ell_a$​, associated with action $a$, such that the loss can be expressed as $\ell(y, a) = \langle y, \ell_a \rangle$. Note that, in the case of classification (i.e. $y$ is one-hot vector indicating the class), $\ell_a$ can be explicitly defined as $\ell_a = (\ell(1, a), \ell(2,a), \ldots, \ell(d, a)]$.
> 5. Decision-Calibration and Decision-Calibration+ReDCal:  In the second paragraph of Section 4.1, we explained that Decision-Calibration corresponds to Algorithm 1, while Decision-Calibration+ReDCal refers to the process of applying ReDCal as a post-processing step after running Decision-Calibration. Including the baseline of Algorithm 1 highlights that using the subroutine alone is suboptimal. By running ReDCal on top of it, we demonstrate that the predictors can be further improved, showcasing the added value of the combined approach.
> 6. In Theorem 3.2, we do not count the number of time steps for Decision-Calibration. In Theorem 3.4, we do count the number of time steps for Decision-Calibration.
> 7. Fixed typo in Definition 2.4, Theorem 3.4 item 1, and $h$ in Algorithm 3. Added a brief description of Reconcile in the paragraph before Theorem 2.7

---

> > ### Comment · Reviewer_ctoE · 2024-11-24
> >
> > Thank you for the updates! Your responses were helpful in clarifying some of my confusions with the paper. I will update my score.

---

### Official Review · Reviewer_eKLK · 2024-11-05

**Soundness:** 3
**Presentation:** 3
**Contribution:** 2
**Rating:** 6
**Confidence:** 3

**Summary:**

This paper studied the problem of model multiplicity in downstream decision-making. In this setting, two predictive models of equivalent accuracy do not agree on their predictions for the downstream decision-making problem.
The paper proposed a new calibration framework which calibrates the predictive models with respect to both a finite set of downstream decision-making problems and the individual probability prediction. Further, the paper proposed an algorithm that first reconciles the differences in individual probability prediction then calibrates the updated models so that they are indistinguishable from the true probability distribution to the decision-makers together with its finite-sample analysis. Numerical experiments on two datasets demonstrate the effectiveness of the proposed method.

**Strengths:**

- This paper studied an important and practical problem of model multiplicity. The paper overall is well-organized and presented with a good clarity.
- It is in particular helpful to have the illustrative example in Figure 1, which directly shows that it is insufficient to only update two predictive models so that they have improved squared loss and nearly agree on their individual predictions almost everywhere.
- Theoretical guarantee shows that the new algorithm ReDCal provides an improved accuracy and approximately agrees on the best-response action almost everywhere compared to the prior work.

**Weaknesses:**

- The experimental results with the HAM10000 dataset show substantially larger error bars, and much less smooth convergence. It is helpful to provide more details on this differences between the two sets of results.
- The experiments only compared to one other baseline proposed in (Roth et al 2023). How does the proposed algorithm compared to other related works in the model multiplicity?

**Questions:**

- W.r.t. the finite sample analysis, is the proposed algorithm robust towards any noise in the samples?

---

> ### Author Response · Authors · 2024-11-22
>
> We would like to thank the reviewer for their time and insightful comments. Please find our responses below:
>
> **Regarding the differences between the two sets of experimental results**: The HAM10000 dataset only contains 10000 samples in total. We split the dataset into train/validation/test sets, with 20% of the data used for validation and 20% of the data used for testing. In comparison, the Imagenet dataset has 40000 samples used for calibration and 10000 samples for testing. Moreover, the downstream decision-making loss function for the HAM10000 experiment is motivated by medical domain knowledge in Zhao et al. (2021) with added Gaussian noise, while the loss function for the Imagenet experiment is randomly generated. The exact details of these differences can be found in the experiment setup in Section 4.1 and Appendix F.2.
>
> **Regarding comparison to other related work in model multiplicity**: We focus on comparing our result with that of Roth et al. 2023 due to the similarity in our approaches to resolving model multiplicity. Moreover, we also compare our empirical results with the baseline algorithm of Decision-Calibration (Algorithm 1).
>
> **Regarding the robustness of our proposed algorithm**: In Section 3.2, we provide a set of results when the decision-makers only have access to a finite dataset instead of the underlying probability distribution. To the best of our knowledge, we are not aware of any work in multicalibration that explicitly discusses robustness toward noise in the samples.

---

> > ### Comment · Reviewer_eKLK · 2024-11-26
> > **update**
> >
> > Thank you for your response! I will keep my original score.

---

### Meta-Review · Area_Chair_cqJh · 2024-12-23

**Metareview:**

This a paper which just sits above the borderline area. The paper studies the problem of model multiplicity in downstream decision-making.
All reviews have been cautiously positive with various suggestions for improving and strengthening the paper. In particular, the paper studies an important problem and provides some theoretical guarantees. I believe the paper is good enough for publication but the authors should incorporate the improvements suggested by the reviewers.

**Additional Comments On Reviewer Discussion:**

A good discussion among reviewers and authors took place which resulted in some scores being updated.

---

### Decision · Program_Chairs · 2025-01-22

Accept (Poster)